# Conserved regulatory switches for the transition from natal down to juvenile feather in birds

Chih-Kuan Chen [1,2,3,4], Yao-Ming Chang [5], Ting-Xin Jiang[1], ZhiCao Yue [6,7,8], Tzu-Yu Liu [1,9], Jiayi Lu[1], Zhou Yu[1], Jinn-Jy Lin [10], Trieu-Duc Vu[11], Tao-Yu Huang[2], Hans I-Chen Harn[1], Chen Siang Ng[3,12,13,14], Ping Wu [1], Cheng-Ming Chuong [1] ✉ & Wen-Hsiung Li [2,15] ✉

The transition from natal downs for heat conservation to juvenile feathers for simple flight is a remarkable environmental adaptation process in avian evolution. However, the underlying epigenetic mechanism for this primary feather transition is mostly unknown. Here we conducted time-ordered gene co-expression network construction, epigenetic analysis, and functional perturbations in developing feather follicles to elucidate four downy-juvenile feather transition events. We report that extracellular matrix reorganization leads to peripheral pulp formation, which mediates epithelial-mesenchymal interactions for branching morphogenesis. α-SMA (*ACTA2*) compartmentalizes dermal papilla stem cells for feather renewal cycling. LEF1 works as a key hub of Wnt signaling to build rachis and converts radial downy to bilateral symmetry. Novel usage of scale keratins strengthens feather sheath with SOX14 as the epigenetic regulator. We show that this primary feather transition is largely conserved in chicken (precocial) and zebra finch (altricial) and discuss the possibility that this evolutionary adaptation process started in feathered dinosaurs.

Evolutionary innovations have enabled birds to occupy different ecological niches. In particular, feathers show a very high degree of diversity, providing an excellent model for studying how animals adapt to different environments[1–4]. Different developmental stages of a bird exhibit different types of feathers. Most hatchling plumages are either naked or have natal downs, which confer heat conservation for the

hatchlings. When juvenile birds are ready to leave the nest, most radially symmetric downs are replaced by bilaterally symmetric juvenile feathers in the same follicle to form basic plumage. After several rounds of molting, the adult feathers that can respond to environmental changes (hormones, seasons, etc.) are eventually formed (Fig. 1a)[5,6]. These remarkable morphological transitions are based on

[1]Department of Pathology, Keck School of Medicine, University of Southern California, Los Angeles, CA, USA. [2]Biodiversity Research Center, Academia Sinica, Taipei, Taiwan. [3]The iEGG and Animal Biotechnology Center, National Chung Hsing University, Taichung, Taiwan. [4]Rong Hsing Research Center for Translational Medicine, National Chung Hsing University, Taichung, Taiwan. [5]Institute of Biomedical Sciences, Academia Sinica, Taipei, Taiwan. [6]Department of Cell Biology and Medical Genetics, Shenzhen University Medical School, Shenzhen, Guangdong, China. [7]International Cancer Center, Shenzhen University Medical School, Shenzhen, Guangdong, China. [8]Guangdong Key Laboratory of Genome Instability and Human Disease Prevention, Shenzhen University Medical School, Shenzhen, Guangdong, China. [9]Department of Life Sciences, National Cheng Kung University, Tainan, Taiwan. [10]National Applied Research Laboratories, National Center for High-performance Computing, Hsinchu, Taiwan. [11]Michigan Neuroscience Institute, University of Michigan School of Medicine, Ann Arbor, MI, USA. [12]Institute of Molecular and Cellular Biology, National Tsing Hua University, Hsinchu, Taiwan. [13]Department of Life Science, National Tsing Hua University, Hsinchu, Taiwan. [14]Bioresource Conservation Research Center, National Tsing Hua University, Hsinchu, Taiwan. [15]Department of Ecology and Evolution, University of Chicago, Chicago, IL, USA. ✉e-mail: cmchuong@usc.edu; whli@uchicago.edu

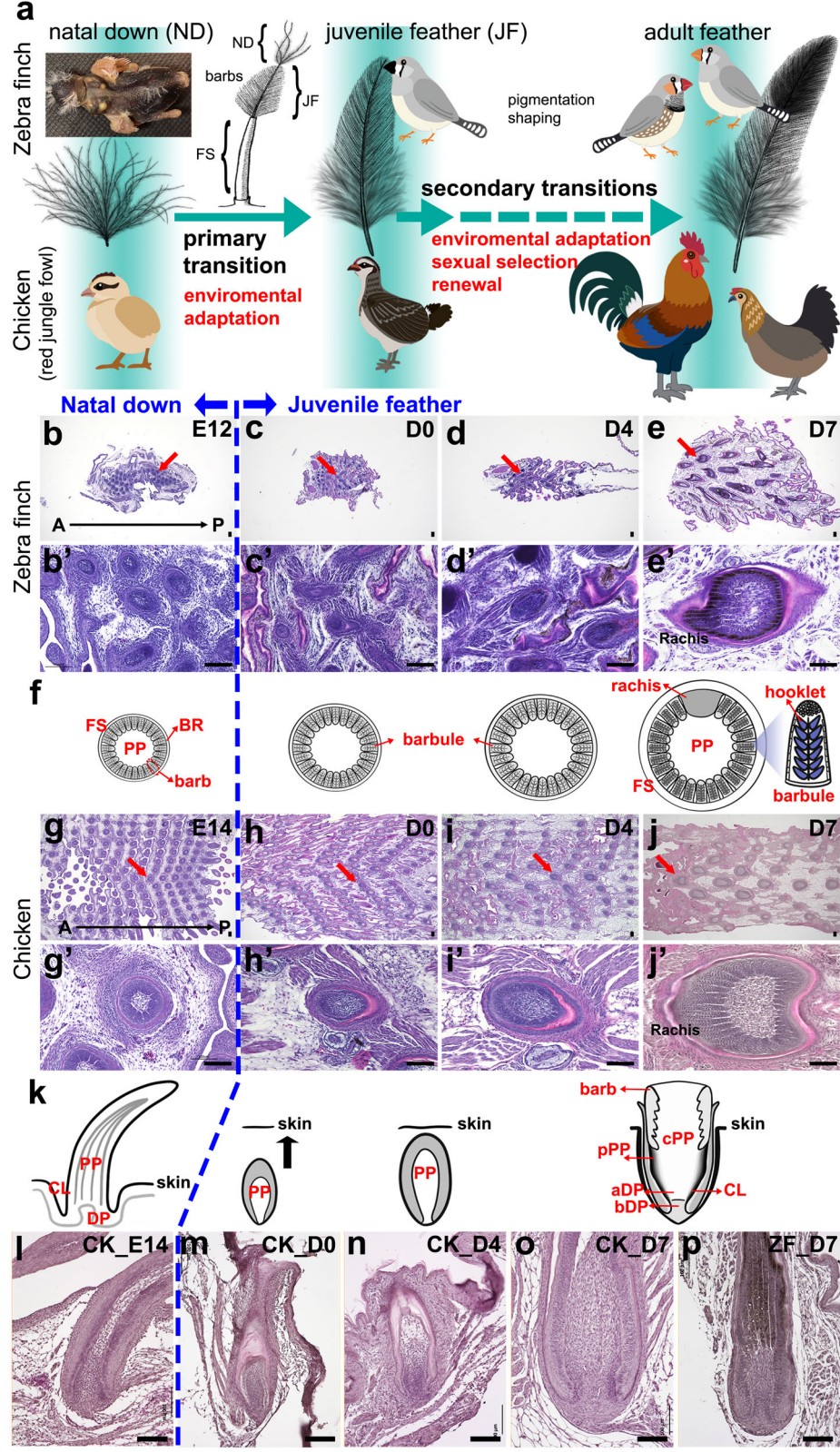

the stem cells and dermal niches from the same feather follicles, and can give plumages different morphologies and functions at different times of a bird's life for optimized function, sometimes giving birds totally different appearances; this process is known as "organ level metamorphosis"[7,8].

The natal down to juvenile (rachidial) feather conversion is here referred to as the primary feather transition process, while the formation of diverse adult feathers after several rounds of molting is referred to as the secondary feather transition process. These two processes set up the foundation to generate regional and timing differences in feather cycling and to achieve optimal functions required at the different stages of adult birds' life[9,10]. The topological changes of feather follicle architectures in natal and adult feathers have been well-characterized[10,11]. The roles of morphogens in forming rachidial and

**Fig. 1 | The morphology and histology of posterior dorsal feathers at different developmental stages of chicken and zebra finch. a** Scheme of the primary feather transitions from natal down (only one barb is shown with barbules) to adult feather in chickens and zebra finches was modified from previous studies[12,17,21]. **b**–**e** H&E stainings of cross sections of the entire posterior dorsal skins in zebra finches (*n* = 5 biologically independent zebra finch skins). A chestnut-flanked white zebra finch mutant was used to visualize the juvenile feathers (Supplementary Fig. 1). **b'**–**e'** A close view of an individual follicle from b-f (indicated by red arrows), respectively. **f** A cross-section view of feather maturation. **g**–**j** H&E stainings of cross sections of the entire posterior dorsal skins in chickens (*n* = 12 biologically independent chicken skins). **g'**–**j'** A close view of an individual follicle from (**g**–**j**) (indicated by red arrows), respectively. **k** A longitudinal section view of the feather maturation. **l**–**p** H&E stainings of longitudinal sections of feather follicles on central posterior dorsal skins. FS feather sheath; JF juvenile follicle; PP pulp; cPP central pulp; pPP peripheral pulp; FF feather filament; DP dermal papilla; BR barb ridge; CL collar; E embryonic incubation days. D posthatch days. A anterior (head); P posterior (tail). Black scale bar: 100 μm.

---

barb branches in adult feathers have also been elucidated[12,13]. The mechanism of secondary transition of feather follicles has been explored in several birds[14–16]. Yet, the evolution and underlying mechanism of the primary transition, which occurs only once in life in most birds, has rarely been studied.

Based on previous studies and our findings, rachis, strong feather sheath, regeneration ability, and feather vane are major distinctions of juvenile feathers from natal downs (Fig. 1a)[6,10,11,17]. The rachis serves as the backbone of the feather, providing support and structural integrity[12,18]. Wnt signaling pathways are the major regulators of both barbule hooklet and rachis development[12,19,20]. Feather sheath is a structure that maintains feather's cylindrical shape until it starts to disintegrate near the tip, allowing the mature part of the feather to unfurl[1]. In adult chickens, feathers need to undergo molting to maintain their normal functions, and the molting process, known as cyclic renewal, is controlled by the dermal papilla and regulated by Wnt inhibitors[21–23]. Feather barbule hooklets are found on the barbules of most bird feathers, which interlock onto the proximal barbules of the immediately adjacent barb and transform the barb branches into a planar vane[12,24]. Here we will focus on the epigenetic controls of the primary feather transition.

To identify signaling pathways and transcription factor (TF) genes involved in the primary feather transition, we obtained embryonic and juvenile time series transcriptomes of chicken posterior dorsal skins and analyzed the data using two approaches. First, we applied the time-ordered gene co-expression network (TO-GCN)[25] to build a kinetic network of the TF genes in the transcriptomes; this TO-GCN can potentially reveal the up-regulation order of TF genes during feather development. Second, avian β-keratin genes are highly repeated in bird genomes and the regulation of their expression has not been well studied[26,27]. Here, we annotated β-keratin genes in the newly released chicken genome (GRCg6a) to analyze the transcriptomes and applied ATAC-seq to elucidate the epigenetic regulation of the specific β-keratin gene subfamilies during the primary feather transition[28,29]. We then functionally validated the predicted molecules in chicken flight feathers. To test the molecular regulatory conservation in birds, we compared the primary feather transitions in chicken (white leghorn) and zebra finch (*Taeniopygia guttata*). Chicken, a precocial bird, belongs to Galliformes, which is a basal lineage of birds. Zebra finch belongs to Passeriformes, which are the most derived birds and the largest avian order; all species in this order are altricial birds[30–32]. A trait conserved in these two distantly related species is likely conserved in most or all birds.

## Results

### The primary feather transition in chicken and zebra finch: from natal downs to juvenile feathers

To study the morphological changes during the primary feather transition in the posterior dorsal skin, we compared the histology of chicken feather follicles between embryos (E12 in zebra finch and E14 in chicken, because the developmental status of E12 zebra finch is close to that of E14 chicken[33]) and hatchlings (post-hatch day 3 to day 7, i.e., D3 to D7), and then compared the structures with that of the posterior dorsal feather follicles in D7 zebra finches. In this region, juvenile feathers grew out from the skin around D6 in both species (Fig. 1a and

Supplementary Figs. 1 and 2), suggesting that the time point of the transition initiation has been conserved in both species.

To understand the morphogenesis of feather follicles during the primary transition, both cross and longitudinal paraffin sections with hematoxylin and eosin (H&E) staining were applied to the whole skins or feather follicles of both species (Fig. 1b–p); the surfaces of the skins were used as the anchor for the comparisons among different stages and species. In cross sections, we found that the structures of the feather germ were similar while the sizes increased slowly from embryos to D4 in both species, suggesting a slow-growing phase (Fig. 1b'–d' and 1g'–i'). The juvenile feather follicles started to enlarge quickly and formed rachis after D5, showing a fast growth phase (Fig. 1b'–e', 1g'–j', and Supplementary Figs. 1 and 2). In D7, the juvenile feathers in chicken and zebra finch showed similar maturities: thickened feather sheathes, rachis and barb ridges were clearly visible (Fig. 1e', f and j'). The clear bard ridges suggest the formation of organized barbs and barbules. This scheme shows the timely dynamics of the primary feather transition in a cross-view (Fig. 1f, from left to right).

To reveal different perspectives, longitudinal sections were conducted in the central (spinal) feather follicles and the results supported the slow-growing phase from embryos to D4 (Fig. 1k and l–n) and the conclusion that the maturity of D7 feather follicles is similar between chicken and zebra finch: curved collars, enlarged pulps, and biconcave dermal papilla are clearly visible (Fig. 1o, p). The peripheral pulp regions show condensed cells, suggesting the formation of peripheral pulps (Fig. 1k, o, p). The scheme showing the time dynamics of the primary feather transition in longitudinal view reveals that juvenile feather germs grow from bottom toward top and eventually protrude out of the skins (Fig. 1k, from left to right). Based on these data and the literature[1,19,21,22], we propose five major morphogenesis events during the primary feather transition in birds: biconcave dermal papilla formation, peripheral pulp formation, rachis formation, vane formation, and feather sheath thickening.

### Coordination of multiple signaling pathways during the primary feather transition

As we wanted to know the overall skin tissue changes during the primary feather transition, we sampled the whole posterior dorsal skins from day 3 to 7 hatchlings in which complex gene interactions from different cell populations and different timings are involved. We applied the time-ordered gene co-expression network (TO-GCN) method[25], which was designed to decipher the molecular regulations from time (or developmental) course transcriptomes of complex tissues, to analyze the transcriptomes from embryonic and juvenile skins. We focused mainly on TF)genes because they are the major players of gene regulation.

RNA-seq analysis of the transcriptomes from embryonic and juvenile samples was conducted to assess the quality of sequencing libraries and generate normalized read counts as the input for the TO-GCN analysis (Supplementary Fig. 3 and Table 1). A TO-GCN of 11 levels was constructed in both feather types based on the expression profiles of the TF genes. The lower (earlier) levels included the TF genes expressed at the earlier feather developmental stages, while the higher (later) levels included those expressed at the later developmental

stages (Fig. 2a, b; Supplementary Data 2). TO-GCN levels represent the timing of TF gene expression. The *d* (= embryonic level − juvenile level) values showed a normal distribution, suggesting that most TF genes are commonly used in both feather types (Fig. 2c; Supplementary Data 2, 3, and 4).

In early embryonic skin development (levels 1 to 5), Rho GTPases and cell cycle-related pathways are enriched (Supplementary Data 3). Rho GTPases are best known for their roles in regulating cytoskeletal rearrangements, cell motility, cell polarity, axon guidance, vesicle trafficking, and the cell cycle[34,35], suggesting that Rho GTPase-mediated cell proliferation and polarization are the major events at this stage. Unexpectedly, polycomb repressive complex 2 (PRC2) related pathways were also enriched. PRC2 is a multiunit epigenetic protein complex that silences gene expression by catalyzing tri-methylation of histone H3 at lysine 27 (H3K27me3)[36]. How methylation is involved in natal down growth regulation is an interesting question. In late embryonic skin samples (levels 6 to 11), cornified envelope and keratin formation pathways were enriched (Supplementary Data 3).

In the juvenile TO-GCN, there are two distinct level groups each with a large gene member, suggesting two morphogenesis groups in post-hatched feathers. The first group is at levels 3 (110 TF genes) and 4 (102 TF genes) in which the major enriched pathways are extracellular matrix (ECM) organization and metabolism (Fig. 2b, e, Supplementary Data 3 and 4). The second group is at levels 9 and 10 in which 72 and 138 TF genes were assigned, respectively. Many molecular pathways were enriched in this period but Wnt/β-catenin related pathways were dominant (Fig. 2b, e, Supplementary Data 3 and 4). ECM is vital for determining and controlling the most fundamental behaviors and characteristics of cells such as proliferation, adhesion, migration, polarity, differentiation, and apoptosis[37,38]. The Wnt/β-catenin related pathway is known to control rachis and barbule hooklet formation[12,19,20]. Both of them are highly associated with the specific structures of juvenile feathers and are therefore the focus in this study.

In addition to ECM and the Wnt/β-catenin related pathways, many other known molecular mechanisms were also enriched. In both early embryonic and juvenile skin development (levels 1 to 5), muscle formation genes were enriched, consistent with the previous finding that muscle development is important for feather positioning (Supplementary Data 3 and 4)[39]. In late juvenile skin development (levels 6 to 11), planar cell polarity (PCP) signaling pathway genes were enriched. A coupling of apical-basal polarity and PCP was identified to interpret the Wnt signaling gradient, which controls the bilateral symmetric feather formation (Supplementary Data 4)[40].

Next, we wanted to know the TF genes that are up-regulated in the primary feather transition. Here we focused on feather type-specific TO-GCN levels from 8 to 11 when the feathers have undergone keratinization, a relatively well-characterized feather developmental stage. For example, if a TF gene is expressed at level 9 of juvenile feather but not at levels 8, 9, 10, or 11 of natal down, it will be defined as a juvenile-specific keratinization TF gene, and vice versa (Fig. 2c). If the TF gene is expressed in both feather types at levels 8, 9, 10, or 11, it will be defined as a common keratinization TF gene (Fig. 2c). We matched the feather type-specific TF genes with those in two previous studies in which keratin regulators were well characterized (Supplementary Data 5)[26,27]. Interestingly, compared to our TF genes identified from embryonic TO-GCN, our TF genes identified from juvenile TO-GCN overlapped more with TF genes controlling scale or claw keratins (embryonic TFs: 0/9 = 0% vs juvenile TFs: 3/8 = 37.5%, Supplementary Data 5), suggesting that scale/claw keratins and their regulators may contribute more to juvenile feathers than to embryonic feathers. To further narrow down the targets, we focused on the TF genes that are significantly upregulated during feather keratinization and eventually obtained 16 juvenile-specific and 6 natal down-specific keratinization TF genes (Fig. 2c). In addition to those overlapped with the two previous studies, some predicted TFs have known functions in skin appendage

development. For example, *GLI1* is transcriptionally regulated by Shh signaling, an important feather activator[3,41]. BNC1 is known to be present in the basal cell layer of the epidermis and in hair follicles. This gene is thought to play a regulatory role in keratinocyte proliferation[42]. ALX4 dysfunction is known to disrupt craniofacial and epidermal development in human[43]. HES5 activity is required for the normal development of the hair cells in the mammalian inner ear[44]. ID3, a BMP target, is expressed in the dermal papilla of both vibrissa and pelage follicles in mouse[45]. To validate the analysis, we performed section in situ hybridization (SISH) of *BNC1* (representing a putative juvenile specific keratinization TF gene) and *ID3* (representing a putative common keratinization TF gene) in both feather types. Indeed, the expression of *BNC1* shows distinct intensities whereas the expression of *ID3* is similar between the two feather types (Fig. 2d).

## ECM re-organization generates peripheral pulp for feather branching morphogenesis

We found that ECM organization was specifically enriched at three levels of the juvenile feather TO-GCN (L2 to L4, Fig. 2e). Components of ECM link together to form a structurally stable composite, contributing to the mechanical properties of tissues. ECM is also a reservoir of growth factors and bioactive molecules[37,38]. In chicken embryonic feathers, ECM regulates mesenchymal mechanics which can spontaneously break skin symmetry[46]. In chicken adult feathers, ECM reorganization enables peripheral pulp formation[21]. Therefore, we asked what is the function of ECM in primary feather transition and which molecules control and maintain the ECM reorganization process.

In the juvenile feather TO-GCN, levels 2 to 4 basically correspond to the D3 to D5 stages in feather development (Fig. 2b). A novel morphogenesis at this stage is the generation of peripheral pulp, which has about five layers of mesenchymal cells closely attached to the feather filament and basement membrane. The peripheral pulp expands the epithelial-mesenchymal interactive interface for barb patterning[21,47]. The gain of ECM-mediated pulp differentiation is therefore our hypothesis for the primary feather transition. Tenascin C (TNC) is frequently used as ECM and differentiated pulp markers[21,22]. The immunochemistry (IHC) signals of TNC reveal that the peripheral pulp was gradually differentiated from embryonic pulp along with the growth of embryonic to juvenile feathers in both chicken and zebra finch (Fig. 3a−e and 3a″−e″). We further picked up two ECM reorganization-related TFs, *TWIST2*, and *ZEB2*, from levels 2 to 4 of TO-GCN to examine their expressions. The SISH of *TWIST2* showed the initial expression in the collar of embryonic feathers, the induction in whole pulp in early juvenile feathers, and the restricted expression in apical dermal papilla and peripheral pulp in late juvenile feathers in both chicken and zebra finch (Fig. 3a1−e1), suggesting its role in peripheral pulp formation. Interestingly, the IHC of ZEB2 showed an almost opposite pattern in that the signals were enriched in the central pup in both species (Fig. 3a2−e2), suggesting that it may maintain the mature tissues.

## α-SMA (*ACTA2*) shapes adult dermal papilla to compartment dermal papilla stem cells for cyclic renewal

Dermal papilla located at the follicle base is essential for cyclic renewal. In our observation, juvenile and adult feathers undergo cyclic renewal but natal downs do not have the regeneration ability. Here we mimic the feather cyclic renewal by juvenile feather plucking and regeneration. *ACTA2* (encoding α-SMA) is a major feather dermal papilla marker and our TO-GCN analysis showed that *ACTA2* co-expressed with levels 2 to 4 TF genes (Supplementary Data 6)[21,22]. We therefore paid additional attention to the role of α-SMA in primary feather transition. The dermal papilla of downy feather is long and slender, while dermal papilla in the juvenile and adult feathers is biconvex-shaped (Fig. 3 left side schemes)[21,22]. We found that, in the IHC on skin longitudinal sections, the expression of α-SMA in dermal papilla increased with the

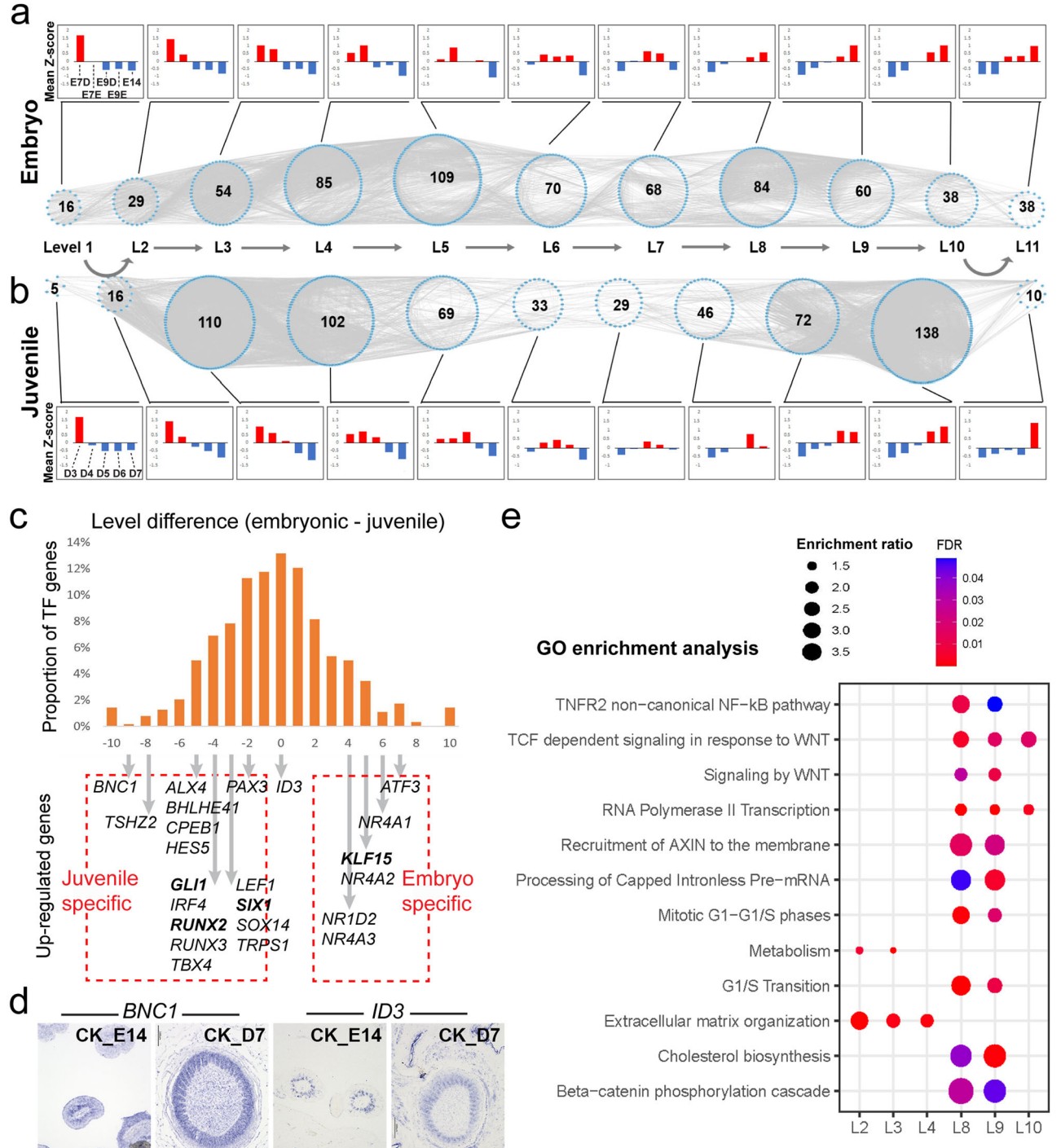

**Fig. 2 | Construction of TO-GCNs and the enriched pathway in the primary feather transition.** The TO-GCNs of TF genes were constructed from normalized gene expression levels during embryonic (**a**) and juvenile feather (**b**) developments. In a TO-GCN, each blue node represents a TF gene and each grey line represents a co-expression relationship between two TF genes. The number in each level represents the number of TF genes at that level. The mean z-score of TF gene expression at each level was used to generate the bar charts and heatmap. The raw input is shown in the Source Data file. **c** The proportion of TF genes in TO-GCN level differences. Most TF genes show less than 3-level differences between embryonic and juvenile feathers. Embryonic (juvenile) specific TFs indicate TFs that are specifically upregulated in levels bigger than 8 in natal downs (juvenile) but not in juvenile (natal downs) feathers. All the specific TFs for natal downs and juvenile feathers are labeled. *ID3* is the control. **d** SISH of *BNC1* shows distinct patterns between the two feather types (barb tips of embryonic feather and whole barbs of juvenile feather) while *ID3* is expressed similarly between the two feather types (*n* = 3 biologically independent chicken skins). **e** The pathways enriched in more than one level were selected from Supplementary Data 4. Epi epidermis; Der dermis; FF feather filament; Skin whole skin; L level of TO-GCN; GO gene ontology; FDR false discovery rate. Scale bar: 100 μm.

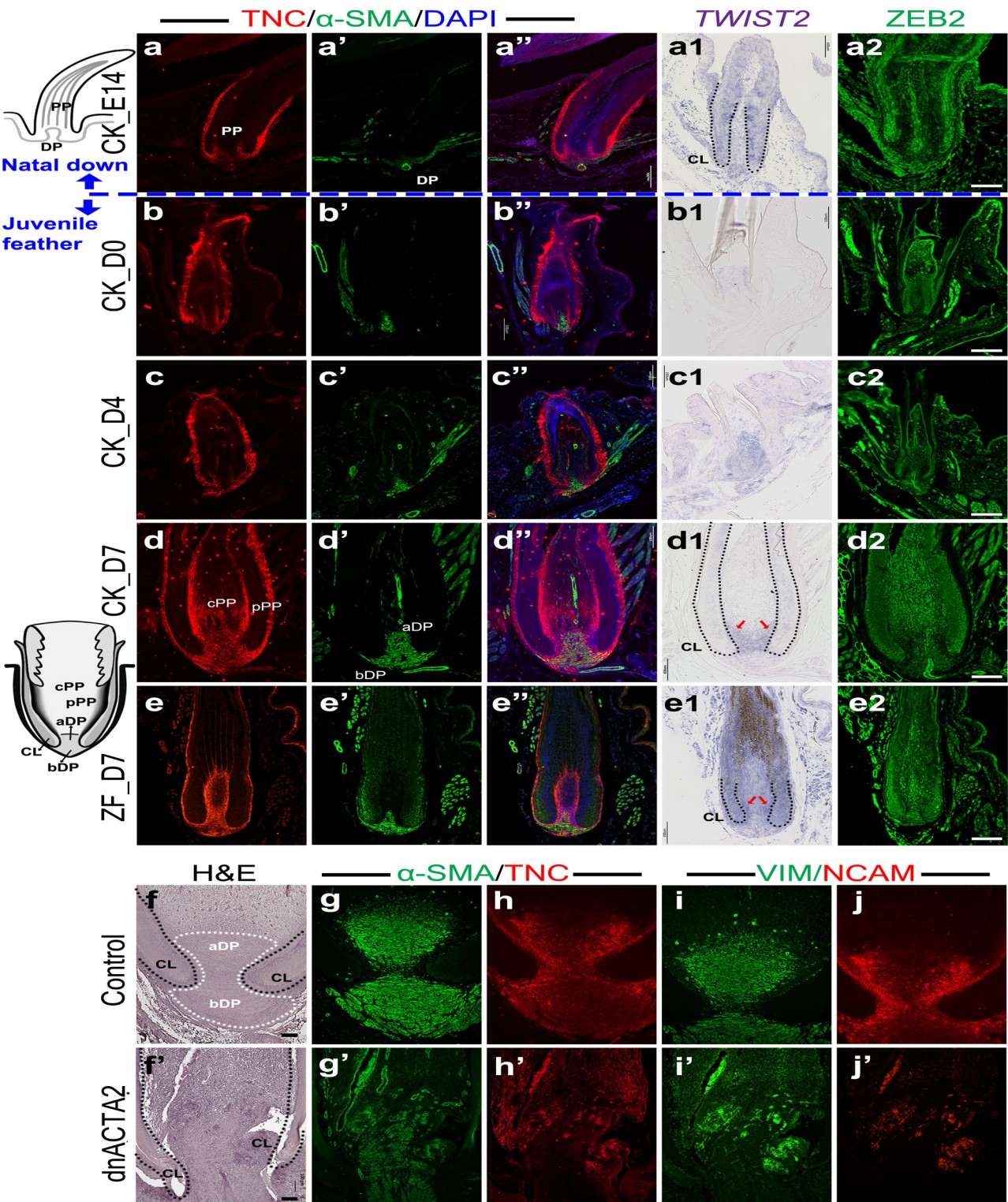

**Fig. 3 | The immunochemistry (IHC) and section in situ hybridization (SISH) of embryonic and juvenile feather follicles in chickens and zebra finches, and the functional perturbation of α-SMA (*ACTA2*) in adult chicken flight feather follicles.** The schemes of embryonic (upper) and juvenile (lower) feather follicles are shown in the left side. The immunostaining of α-SMA (*ACTA2*), TNC, and DAPI in embryonic chicken (**a**–**a″**), juvenile chicken (**b**–**d″**), and juvenile zebra finch (**e**–**e″**) feather follicles (*n* = 5 biologically independent samples). **a1**–**e1** The SISH with *TWIST2* in the same tissues with (**a**–**e″**). The distinguishable collar regions are highlighted by black dot lines and the signals enriched in the peripheral pulps are

indicated by red arrows (*n* = 3 biologically independent samples). **a2**–**e2** The IHC with ZEB2 in the same tissues with (**a**–**e″**). H&E staining (**f**) and immunostaining of α-SMA (**g**), TNC (**h**), VIM (**i**), and NCAM (**j**) in control follicles. (*n* = 2 biologically independent samples). H&E (**f′**) staining and immunostaining of α-SMA (**g′**), TNC (**h′**), VIM (**i′**), and NCAM (**j′**) in RCAS-dnACTA2 injected follicles (*n* = 7 biologically independent samples). CK chicken; ZF zebra finch; PP pulp; pPP peripheral pulp; cPP central pulp; DP dermal papilla; aDP apical dermal papilla; bDP basal dermal papilla; CL collar epidermis. Scale bar: 100 μm.

juvenile follicle development and eventually filled up the biconvex-shaped dermal papilla in D7 juvenile feathers in both chicken and zebra finch (Fig. 3a'–e'). Most interestingly, the study of feather renewal cycling showed that in the resting phase, dermal papilla stem cells are located in the apical part of the bi-concaved dermal papilla, suggesting that the biconvex-shaped structure is essential for feather cyclic renewal[21]. In the growth phase, the activation of apical part dermal papilla generates pulp progenitors, giving rise to both central pulp for nutrition purposes and peripheral pulp for continuous epithelial-mesenchymal interactions required for feather branching morphogenesis.

We then tested the importance of *ACTA2* in dermal papilla formation. Since the protein sequence of *ACTA2* is identical between humans and chickens, we cloned a published human dominant-negative *ACTA2* form into RCAS virus (RCAS-dnACTA2) and injected it into the cavities of the plucked flight feather in chicken[48]. The knock-down virus caused such severe effects that most (7/8) of the injected follicles could only regenerate tiny or no visible feathers two weeks after injection (Supplementary Fig. 4a–d). All eight injected follicles could not be regenerated after the second plucking. Histologically, the dermal papilla, part of the collar structures, and the barb structures were also disrupted (Fig. 3f and f', Supplementary Fig. 4d), suggesting that the establishment of organized dermal papilla is the most essential step for feather cyclic renewal. To further understand the molecular changes in those abnormal follicles, immunostaining of several known ECM factors as well as α-SMA (*ACTA2*) were applied to feather sections (Fig. 3g–j')[21]. Dermal papilla markers α-SMA and vimentin (VIM) were enriched in whole dermal papilla in the control feather follicles but attenuated and dispersed in the virus-injected follicles (Fig. 3g and g', i and i'). Unexpectedly, TNC and Neural Cell Adhesion Molecule (NCAM), which were found to be located in papilla ectoderm in the control feather follicles, were attenuated and dispersed in the virus-injected follicles (Fig. 3h and h', j and j'), suggesting the simultaneous disruption of dermal papilla and pulp structures. It could be that the dermal papilla of juvenile feathers contains myofibroblast cells and knock-down of α-SMA may then affect the integrity of these cells. These results suggest that α-SMA could serve as a structural component to build the micro-environment for the primary feather transition in birds".

### Wnt gradient is the major regulator of rachis formation and LEF1 is a key molecular hub converting radial downy to bilaterally symmetric juvenile feathers

The molecular gradient in feather follicles from anterior to posterior end is crucial for regulating the angles of barb ridges for rachis formation[19,41]. During rachis formation, the anterior to posterior Wnt3a gradient is known to convert radial to bilateral feather symmetry via convergence of barb ridges toward the rachis region[19]. Moreover, multiple Wnt genes showed gradient expressions during flight feather regeneration[23], suggesting that the Wnt-based regulation could be redundant or region-specific.

In the RNA-seq analysis, the expression levels of *Wnt2b*, *Wnt5a*, *Wnt5b*, *Wnt7a*, *Wnt9a*, *Wnt9b*, and *Wnt16* were increased with the juvenile dorsal feather development, corresponding to the time point of the rachis formation (Figs. 1g'–j' and 4a). *Wnt3a* showed constant expression and might not be a rachis regulator of the juvenile dorsal feathers (Fig. 4a). In our TO-GCN analysis, the lymphoid enhancer-binding factor 1 (*LEF1*) was identified as a level 10 TF in juvenile feather development and is known to be a key mediator of Wnt signaling in diverse biological processes (Fig. 4a and Supplementary Data 2)[49,50]. In many cases, it interacts with β-catenin (encoded by *CTNNB1*) to achieve the Wnt signaling regulation and *CTNNB1* is also an important feather growth factor[49,51,52]. Previous studies revealed that rachidial (pennaceous) feather structure evolved through the integration of barb ridge morphogenesis with a second, local inhibitor and an anterior-posterior

signal gradient within the feather[19,41]. The SISH of *LEF1* and *CTNNB1* in embryonic skins show distinct patterns between chicken and zebra finch (Fig. 4b). Moreover, in D7 chicken posterior dorsal skins *LEF1* was expressed in an anterior to posterior gradient in the feather epidermis but *CTNNB1* was expressed evenly (Fig. 4b). These findings suggest that LEF1 is the key factor responding to Wnt signaling. In zebra finch, although *LEF1* was also expressed in juvenile feather epidermis, the feather pigments prevented us from visualizing the gradient (Fig. 4b, lower panel).

To validate the effect of LEF1 on rachis formation, a dominant negative form of LEF1 (dnLEF1) was cloned into RCAS virus and injected into cavities of the plucked flight feathers. Previous studies showed that in flight feathers, the misexpression of Wnt inhibitor DKK1 could slightly disturb the rachis formation, and the overexpression of Wnt3a could disrupt the rachis formation and also cause abnormal barbs[19]. Here, a high proportion of the flight feathers injected with RCAS-dnLEF1 showed defects in rachis (20 of 30, Supplementary Fig. 4e-g). Half of the defective feathers lost part of the rachis while the others lost the entire rachis without influencing the surrounding barbs (Fig. 4c, Supplementary Fig. 4f), suggesting that the function of LEF1 is specific in feather follicles. SHH is a key morphogen for barb formation via their expression in bard ridges and absence in rachis forming regions[19,41,53]. In rachidial feather, barb ridges insert into the rachidial ridge with the helical insertion angle (Fig. 4d, indicated by θ). However, in both natal down and RCAS-dnLEF1 misexpressed flight feather, all the barb ridges were formed in nearly parallel (Fig. 4d and Supplementary Fig. 4h), suggesting that barb ridges were not able to insert into rachidial ridge. Interestingly, the value of θ is in proportion to the size of the rachis (Supplementary Fig. 4h), corresponding to a previous finding[54].

### Many scale keratins are specifically upregulated in juvenile feather sheath by SOX14

β-keratin genes are mainly classified into feather, scale, claw keratins, and keratinocytes[55,56]. A previous study revealed that chicken dorsal natal downs mainly express feather-β-keratin genes on Chr1, Chr10, Chr25, and also some members on Chr27 during keratinization, whereas feather-β-keratin genes on Chr2 and Chr6 are exclusively enriched in adult wing feathers[55]. However, subsequent studies revealed that β-keratin genes are differentially regulated in different skin regions[26,27,56]. Since our TO-GCN analysis along with the previous studies suggested the importance of scale keratins in the juvenile feather formation[26,27], we employed several modifications in the transcriptomic analysis to decipher the β-keratin gene regulation: (1) Our embryonic and juvenile feather tissues were only from chicken posterior dorsal skins. (2) In addition to β-keratin genes, we analyzed the whole epidermal differentiation complex (EDC), which was known to participate in feather functional evolution[57–59]. (3) We used the newly published chicken genome (GRCg6a), which has a much-improved assembly in microchromosomes, where EDC gene clusters are located. (4) We manually annotated the EDC genes, especially the β-keratin genes, which are still poorly annotated in GRCg6a. (5) We conducted an ATAC-seq analysis of embryonic and adult feather tissues to look for epigenetic regulators during the primary feather transition.

We analyzed the co-expression profile of EDC genes in both feather types during their development. Our data were partially inconsistent with the previous finding, showing that, regardless of chromosomal locations, most of the β-keratin genes were expressed during both embryonic and juvenile feather keratinization (Clusters 7 and 11, Fig. 5a). Genes in cluster 11 were highly expressed at the keratinizing stage in both feather types, which include most feather β-keratin genes located on Chr1, 2, 7, 10, 25, and 27 (Fig. 5a; Supplementary Data 7). Genes in Cluster 7 are highly expressed at both the keratinizing stage and a stage ahead of it in both feather types, which include other β-keratin genes, such as scale, claw, feather-like

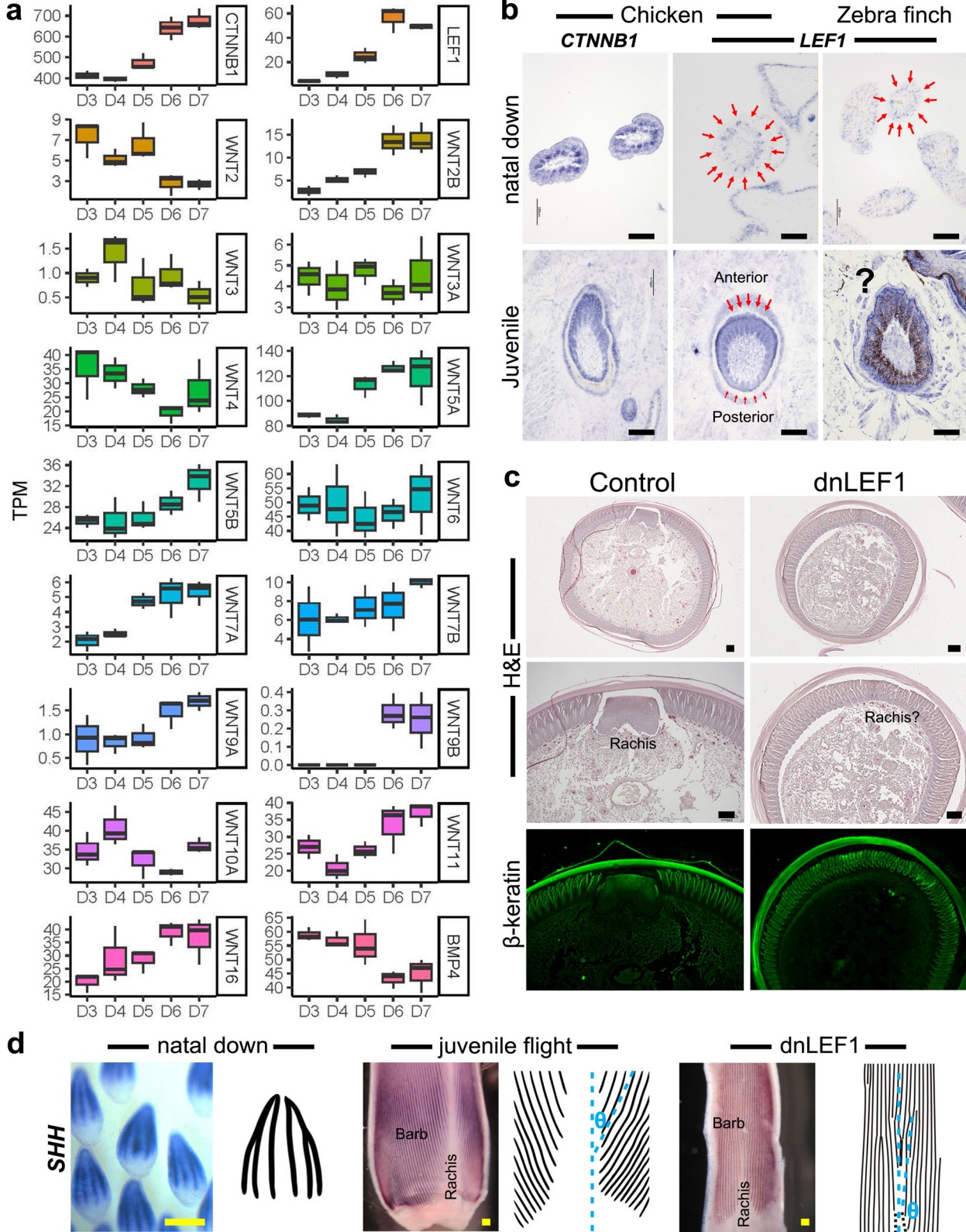

β-keratin, and feather-β-keratin genes on Chr10 (Fig. 5a; Supplementary Data 7). In addition to β-keratin genes, most other EDC genes also were in cluster 7 or 11 (Fig. 5a; Supplementary Data 7).

Only small co-expression clusters and individual genes showed distinct profiles between the two feather types. Two β-keratin genes in Clusters 8 and 9 (*GG6AChr25Ktn11* and *GG6AChr25FK5*) were highly expressed in keratinizing juvenile feathers but not in natal downs

(Fig. 5a; Supplementary Data 7). In Cluster 7, *CRNN* was highly expressed in embryonic but not in juvenile feathers (Fig. 5a; Supplementary Data 7). Interestingly, when we analyzed the β-keratin genes based on their subfamilies (β-keratin related proteins, feather keratins, feather keratin-like proteins, scale keratins, claw keratins, and keratinocytes), β-keratin gene expression differences between the two feather types could be found. Chr25 feather and feather-like keratin

**Fig. 4 | The expression profiles of Wnt signaling genes and the results of functional peturbation. a** The expression profiles of *Wnts*, *CTNNB1*, and *LEF1* during the development of chicken posterior dorsal skins from D3 to D7 (*n* = 3 biologically independent chicken skins over 3 independent samplings). The inter-quartile range (IQR) of boxplot is between Q1 and Q3 and center line indicates median value. Whiskers of boxplot are extended to the maxima and minima. Maxima is Q3 + 1.5 × IQR and minima is Q1 − 1.5 × IQR. The raw data are shown in the Source Data file. **b** The *CTNNB1* and *LEF1* expression patterns in posterior dorsal skins of embryonic and D7 chicken (*n* = 4 biologically independent samples) and zebra finch (*n* = 3 biologically independent samples). *LEF1* shows similar expression pattern in the natal down of the two species (indicated by red arrows). In juvenile feathers, *LEF1* shows the expression gradient from anterior toward posterior direction (the expression intensities are indicated by length of the red arrows) but

*CTNNB1* does not. The question mark indicates that the expression profile is shielded by pigments and cannot be clearly viewed. **c** First panel: the H&E staning of the cross section of feather follicle with RCAS-dnLEF1 misexpression (*n* = 6 biologically independent samples). Second panel: the enlargement of the rachis area in the first panel. Third panel: the β-keratin immunostaining of the second panel to indicate the structural integrity of the barbs. **d** WISH with *SHH* in natal down follicle (*n* = 3 biologically independent samples), juvenile flight feather follicle (*n* = 4 biologically independent samples), and juvenile flight feather follicle with RCAS-dnLEF1 misexpressions (*n* = 20 biologically independent samples). θ indicates the angle between rachis and barbs. In normal juvenile flight feather the θ values is 9.27° while in RCAS-dnLEF1 misexpressed feather follicle the θ values is 7°. Scale bar: 100 μm.

genes were expressed higher in natal downs than in juvenile feathers, while Chr25 scale keratin genes were expressed higher in juvenile feathers (Fig. 5a, b, Supplementary Fig. 5a, Supplementary Data 7), suggesting that: (1) the ratio differences of Chr25 β-keratin gene expression in different chromosomes may contribute to feather type differences, and (2) some scale keratin genes could be important specifically for juvenile feather formation. To understand the expression differences of the scale keratin genes in the two feather types, we conducted SISH with *GG6AChr25Scale2* and *GG6AChr25Scale10* in cross-sections of embryonic and juvenile feather follicles (Fig. 5c–f and Supplementary Data 7). Although the two keratin genes were faintly expressed in the barbs of both feather types, they were highly enriched in the feather sheath in juvenile feathers but not in natal downs.

What TFs control the scale keratins and whether this regulation is conserved in birds were the next questions. All the scale keratin genes are on chromosome 25 for both chicken and zebra finch. We first compared the synteny of chromosome 25 between the two species and found that they are basically conserved along the whole chromosome. Next, we conducted ATAC-seq to compare the differential genomic accessible regions between embryonic feather filaments and adult flight feather follicles. Figure 5g shows the ATAC signals of the two tissues surrounding scale keratin gene cluster. The differential accessible regions overlapped with the conserved chromosome 25 sequences were extracted for footprint analysis to detect the enriched binding motifs and TFs. The results revealed that TFs SOX14, ESRRB, ESRRG, PRDM4, SREBF2, and SMAD5 were enriched in adult flight feathers; the corresponding binding motifs are shown in Fig. 5g'. ESRRB and ESRRG are estrogen-related receptors (GeneCards). Since we did not separate male and female tissues, this prediction could be from sampling bias. However, it also indicates that β-keratin genes could be regulated by hormones. PRDM4 (PR domain zinc finger protein 4) is a transcription factor that plays key roles in stem cell self-renewal and tumorigenesis. The down-regulation of *PRDM4* during juvenile feather growth suggests its negative regulation to keratinization (Supplementary Data 2)[60]. SMADs are a group of signaling mediators and antagonists of the transforming growth factor-beta (TGF-beta) superfamily. SMAD4 and SMAD7 are important for hair follicle development and differentiation[61], but the function of SMAD5 in keratinization is not clear. SREBF2 (Sterol regulatory element-binding protein 2) is key transcription factor that regulates expression of genes involved in cholesterol biosynthesis[62]. The SISH of *SREBF2* shows strong expressions inside the feather sheaths and barbs of both embryonic and juvenile feathers (Supplementary Fig. 5b), suggesting its regulatory role in keratinization in these regions.

Since SOX14 is the most significant TF and greater expressed in juvenile feathers than in embryonic feather transcriptomes (Fig. 5g and Supplementary Data 1), we used SISH to detect its expression in the natal and juvenile feather follicles of chickens and zebra finches. The results showed that the same as the scale keratin genes, *SOX14* was detected in barbs of both feather types in both species, but the expression was only enriched in the feather sheath of juvenile follicles

in both species (Fig. 5h–k). The overlapped expression of scale keratin genes and *SOX14* suggests that SOX14 is an upstream transcription factor of the scale keratin genes. To functionally validate the prediction, we overexpressed SOX14 in embryonic feather using the RCAS-virus system and 1/3 of the embryos (*N* = 12) generated abnormal feather follicles with either round or thickened and shortened phenotypes (Fig. 5l). Since virus infection is partial and the same skin region can include both abnormal and normal feather follicles, SISH was applied to detect the expression of *SOX14* and scale keratin genes in the same skin sections. Follicles with higher *SOX14* expressions showed thickened feather sheath and enhanced expressions of *GG6AChr25Scale2* and *GG6AChr25Scale10* (Fig. 5l and Supplementary Fig. 5c), suggesting that SOX14 is the regulator of the scale keratins in feather sheath.

## Discussion

Feather diversity appears in different body regions and at different developmental stages of a bird[17]. Previous studies revealed regulatory differences among different body feathers or among different parts of a chicken feather[13,26,27]. Feather transition represents a novel type of timing control evolved for better adaptation, depending on different needs at different lifetimes, but the underlying molecular controls have not been well characterized. The primary feather transition is achieved by five tissue reorganizations and our study revealed four of them: 1. Wnt is the common signaling for rachis formation and LEF1 is the downstream key hub for diverse Wnt proteins. 2. ECM reorganization is essential for peripheral pulp formation for further branching morphogenesis. 3. α-SMA (*ACTA2*) is a key factor to compartment dermal papilla formation for feather cycling. 4. Scale keratin is recruited from scale to strengthen the sheath of juvenile feathers and SOX14 appears to be a major activator for this co-option process. Feather vane formation achieved by barbule hooklets is a primary feather transition step that was not included in this study. Although our transcriptomes included the initial time points for barbule hooklet development (D7), we did not identify regulators of the known morphogen WNT2B[12]. Since barbule hooklets contribute to a feather fraction, the finer analysis may be worth pursuing in the future. Note that barbule hooklets have been lost in most ratites[63]. The morphogenesis events and the underlying molecular regulators during the primary feather transition are summarized in Fig. 6. All the chicken primary feather transition factors showed the same expression patterns in zebra finch feather follicles, suggesting that the primary feather transition has been conserved in birds.

The five molecular events basically appear in different developmental time frames. How the sequential morphogenesis is established is an interesting question. Whether successive morphogenetic changes interact with each other is another interesting issue. It seems that the establishment of organized dermal papilla is the most basic step because the overexpression of dnACTA2 not only stops the feather regeneration but also disables the barb formation (Supplementary Fig. 4d), and thus the rachis and barbule hooklets fail to form. The only

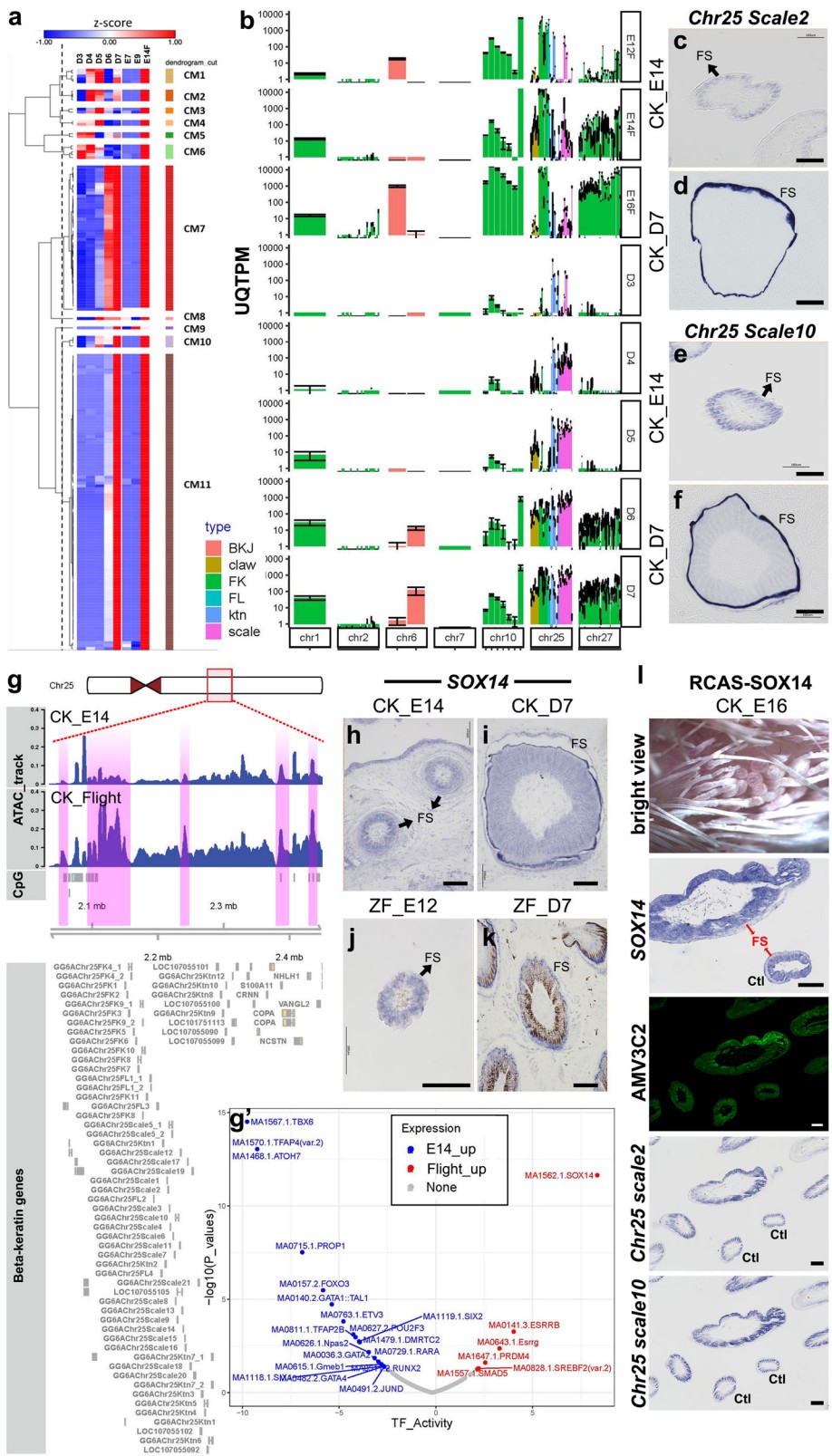

remaining growing structure is the feather sheath, suggesting that feather sheath formation, or follicle exterior morphogenesis, is an independent event. Moreover, although dnLEF1 suppresses rachis formation, IHC of β-keratin in dnLEF1 overexpressed follicles shows similar barbule pattern with that in control follicles (Fig. 4), suggesting that rachis formation and barbule hooklet formation are two independent events.

Multiple Wnt genes seem to participate in the primary feather transition. Whether it is simply functional redundancy or regional specificity is an interesting question. Hox genes represent the best example that elicits distinct developmental programs along the head-to-tail axis of animals and are the upstream signals of many Wnt proteins[64–66]. Whether distinct HOXs activate different WNTs and generate regionally specific feather rachis is another interesting

**Fig. 5 | Expression profiles of EDC genes during the primary feather transition and validaion of SOX14 as the epigenetic regulator of scale keratins in the feather sheath. a** Clustering analysis and the heatmap of chicken EDC gene clusters. The raw data are shown in the Source Data file. **b** Expression of EDC gene clusters in different chromosomes in chicken (*n* = 3 biologically independent chicken skins over 3 independent samplings). The error bar is given as SD. The raw data are shown in the Source Data file. **c–f** In situ hybridizations of *scale keratin 2* and *scale keratin 10* in a cross-section of E14 (CK_E14) and D7 (CK_D7) chicken feather follicles. (*n* = 3 biologically independent samples). **g** Visualization of ATAC peaks of chicken E14 feathers and adult flight feather follicles surrounding the scale keratin cluster. The stronger ATAC peaks in the flight feather follicles are highlighted in pink. **g′** Footprint analysis of differential ATAC peaks between E14 feathers and adult flight feather follicles. The enriched binding motif and the transcription factors in either E14 and adult flight samples were labeled.

Comparisons based on non-paired distributions were performed with the Wilcoxon rank sum test. The reported p values were corrected using the Benjamini-Hochberg method. The raw data are shown in the Source Data file. **h–k** In situ hybridizations of *SOX14* in cross sections of D7 chicken (CKD7) and zebra finch (ZFD7) feather follicles (*n* = 5 biologically independent samples). The feather sheathes are indicated by black arrows. **l** The functional validation of SOX14 overexpression using RCAS virus in the embryonic chicken skin (*n* = 6 biologically independent samples). In the *SOX14* upregulated follicles, the feather sheath, indicated by red arrows, is enlarged. In the same follicles, the expression of RCAS gag protein (AMV3C2), *scale keratin 2,* and *scale keratin 10* are increased. CM cluster module; UQTPM upper quartile TPM; BKJ beta-keratin-related protein; claw claw keratin; FK feather keratin; FL feather keratin-like protein; ktn keratinocyte; scale scale keratin; E12F E12 feather filament; FS feather sheath; Ctl control feather follicle. Scale bar: 100 μm.

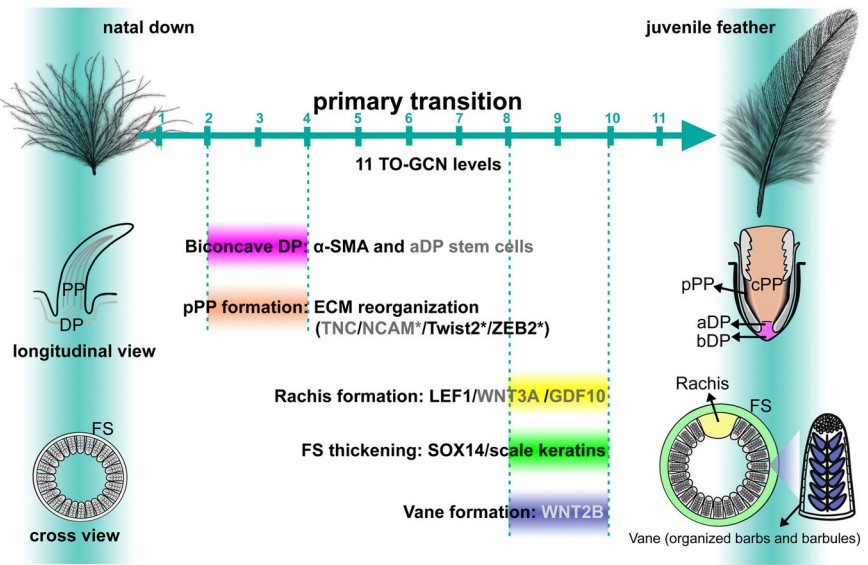

**Fig. 6 | Summary of the phenotypic and molecular changes during the primary feather transition in birds.** Five evolutionary morphogenesis events are shown with their timings indicated by TO-GCN levels. The identified genes for the primary feather transition are listed after the events and shown in either black font color, which is identified in this study, or gray color, which was identified in previous studies[12,19–22]. Peripheral pulp and apical dermal papilla formations for feather regeneration are the early major events that occur from TO-GCN level 2 to 4, while rachis, thickened feather sheath, and vane for multi-functional structures are the late major events that occur between TO-GCN level 8 and 10. PP pulp; pPP peripheral pulp; cPP central pulp; DP dermal papilla; aDP apical dermal papilla; bDP basal dermal papilla; FS feather sheath.

question. In feather regeneration, our previous study has uncovered a cell movement from apical dermal papilla to peripheral pulp along the collar bulge[21]. This remarkable feature allows prolonged interactions between the dermal niche and epidermal stem cells, providing a tunable interface that is essential for barb patterning[21]. The path of the cell movement corresponds to the position of ECM molecule expression, and so ECM reorganization could be a key feather regeneration factor. *ACTA2* is a dermal papilla marker but is not known to regulate ECM molecules like TNC and NCAM. While the knockdown of *ACTA2* disrupted their expressions, we proposed that α-SMA (*ACTA2*) could be used to establish a competent environment for molecular interaction during peripheral pulp and apical dermal papilla formation. Interestingly, we identified *TWIST2*, whose expression during feather transition corresponds to the position of apical dermal papilla and peripheral pulp, suggesting its possible role in feather regeneration. TWIST2 is known to regulate ECM in some cases[67,68].

Previous studies have indicated that SOX genes are important for scale keratin regulation and overexpression of SOX18 enhanced the expression of a scale keratin in embryonic feather[26,27,69]. However, different SOX genes were identified as β-keratin regulators in different skin appendages (SOX10 in scale, SOX18 in embryonic feather sheath, and SOX14 in juvenile feather sheath in this study). It implies that SOX

genes perform regional or temporal specific regulations. Moreover, based on the differential *SOX14* regulation and ATAC-seq analysis, we hypothesize that the embryonic feather sheath can be suppressed epigenetically (Fig. 5). The primary feather transition activates the juvenile feather sheath cells to recruit SOX14 to trigger the expression of β-keratins to enhance keratinization (Fig. 6). Notably, a juvenile bird shows various feather types[17]. Although we focus on the major one, i.e., the contour feathers with vane structure, whether other feather types, like bristles and filoplumes, are achieved by the same molecular mechanisms is another interesting question.

Precocial chickens are covered with natal downs, while altricial zebra finches have only limited natal downs in hatchlings[31,32] (Supplementary Figs. 1 and 2). Despite these differences between chicken and zebra finch hatchlings, their juvenile feather developmental profiles are quite similar[30] (Fig. 1). After several rounds of molting, diverse feathers are derived from the juvenile feather follicles in the secondary transition process, as seen in the sail-feathers of the mandarin duck (*Aix galericulata*) or colorful contour feather of the pheasants[70,71]. The conserved primary feather transition suggests its functional importance for survival while the following diverse feather transitions are mainly for mating choices. Interestingly, although egg incubation in chickens is around 21 days while that in zebra finches is around only

14 days, their juvenile feathers are visible at similar post-hatch stages (Fig. 1e' and j', Supplementary Figs. 1 and 2), suggesting that the primary feather transition is sensitive to the post-hatch stimulus, but are regardless of nesting habitats and parental cares.

The conserved primary feather transition among birds implies conserved regulatory sequences in their genomes. How to identify the regulatory regions and how the regulations are controlled are important questions. Although many chicken mutants have been created, the phenotype without the primary feather transition has never been found[72], suggesting that the loss of feather transition is highly detrimental. The avian conserved non-exonic elements (CNEEs) database could be used to screen the regulatory sequences and the combination of epigenetic marks can be used to understand the epigenetic changes of the regulatory sequences[73]. On the other hand, when the mechanism evolved is a fascinating question. The evolution of feather structures in Dinosauria was proposed in the previous studies in which the basal family Compsognathidae (such as *Sinosauropteryx*) was covered with unbranched monofilaments (down feather without barbules) while the derived family Caudipteridae (such as *Caudipteryx*) showed feather types similar to current birds[74,75]. Whether Dinosauria encompassed feather transitions during their development is not clear, but both the natal and juvenile feather forms have been discovered, suggesting that the primary feather transition in current birds could either be inherited or modified from their dinosaur ancestry.

The success of the Aves in venturing into new eco-spaces relied heavily on the flexibility to convert their feather phenotypes during the lifetime of an individual, which is based on the module-based complexity formation[9]. These modules are feather follicles consisting of stem cells and their niches. In this study, we tried to decipher the strategies used to evolve these regulatory switches in these modules. We found no new molecules needed to be created. Instead, the transcription regulation redeployed existing molecular module and cellular components, which include the topological positioning of the Wnt signaling center, the tissue remodeling to extend the interface for epidermal-dermal interactions, the re-shaping dermal papilla to create dermal papilla stem cell compartment and the co-optional use of scale keratins in feather sheath. Conversions of integument phenotypes from the young to the adult are also seen in some mammals and other animals. These transitions may have been driven by environmental adaptation[76]. Some strategies we learn here may help us understand this fundamental mechanism in Evo-Devo.

## Methods

### Ethics statement
All the animals used in this study were processed following the approved protocols of the Institutional Animal Care and Use Committees of the University of Southern California (USC; Los Angeles, CA, USA) and the Institutional Animal Care and Use Committees of National Chung Hsing University (NCHU, Taichung, Taiwan).

### Sample collection
Fertilized pathogen-free White Leghorn chicken eggs were purchased from Charles River Laboratories. Eggs were incubated at 100°F and 65% humidity until embryos reached the desired developmental stages. Chicken embryos at 14 days (E14), posthatched 3 days (D3), posthatched 4 days (D4), posthatched 5 days (D5), posthatched 6 days (D6), and posthatched 7 days (D7) were used. Chicken hatchlings are mostly covered by natal downs. To visualize the juvenile feather growth, we plucked posterior dorsal natal downs of the hatched chicken and observed the growth patterns of juvenile feathers (Fig. 1). Although a previous mammalian study found that hair plucking can stimulate early regeneration of underneath hairs[77], this phenomenon is not observed in the primary feather transition because the plucked and un-plucked region showed similar juvenile feather growth in both species (Fig. 1 and Supplementary Figs. 1 and 2). The zebra finch eggs

were hatched and the hatchlings were raised in the bird breeding room at NCHU. E12 and D7 Zebra finches were used.

### Paraffin sections and immunostaining
The chicken and zebra finch posterior dorsal skins were dissected and fixed in 4% paraformaldehyde (PFA) at 4 °C overnight, and 7 μm paraffin sections were obtained. Hematoxylin and eosin (H&E) staining and immunostaining were performed following a previous procedure[56,78]. Specifically, after 6% $H_2O_2$ treatment for 10 min, the primary antibody was added to the slides and incubated overnight at 4 °C with agitation. The samples were washed with TBST (Tris Buffered Saline Tween 20) and the secondary antibody was added for 1 h at room temperature. The following antibodies were used in the immunostaining. Primary antibodies: α-SMA (Invitrogen, MA1-06110, 1:50), vimentin (Developmental Studies Hybridoma Bank, H5, 1:30), ZEB2 (Proteintech, 14026-1-AP, 1:50). NCAM and TNC were made from the Chuong lab (1:100)[79]; secondary antibodies: Alexa Fluor secondary antibodies (Invitrogen, 1:1000) were used for fluorescence detection. DAPI (1:2000) was used to visualize the nuclei.

### Quantitative PCR
To quantify the candidate gene expressions, the cDNAs were synthesized from the total RNA by QuaniTect Reverse Transcription kit (Qiagen). Each cDNA sample containing SYBR green (KAPA SYBR FAST qPCR kit) was run on LightCycler 480 (Roche) under the appropriate conditions. Quantification of the TATA box binding protein (*TBP*) RNA was used to normalize target gene expression levels. *SHH* forward primer: CTGGTGAAGGACCTGAGCCCT; reverse primer: GCCCAACT GTGCTCCTCGAT, *TBP* forward primer: CACAGCAAGCGACACAGGGA; reverse primer: AGGTGTGGTTCCCGGCAAAG. Feather keratin forward primer: CAGGAAGGGGCAATCCCGTG; reverse primer: TGAGGAGCCT CGTAGCCCAT. The raw data are shown in Source Data file.

### Collection and construction of the transcriptomic libraries
To choose the appropriate time points representing juvenile feather development, we screened several feather morphogens and found that *SHH* is a good marker gene because its expression corresponded to the phenotypic changes of the juvenile feather follicles (Fig. 1 and Supplementary Fig. 3a). Based on the expression profile of *SHH*, we selected posthatch day 3 to 7 posterior dorsal skins of the chickens to represent the primary feather transition. All the chicken transcriptomic libraries from the posterior dorsal area used in this study and their sources are listed in Supplementary Table 1. For posthatch samples, we collected the whole skin because the feather follicles cannot be quickly dissected to ensure the RNA quality at these stages. The dissected tissues were preserved in RNAlater solution (Invitrogen), incubated at 4 °C overnight, and then transferred to −20 °C until processing for isolation of total RNA. Total RNAs were isolated using the RNeasy Fibrous Tissue Mini Kit (Qiagen). The 30 min DNaseI treatment indicated by the kit was carried out at room temperature. The RNA quantities and qualities of each individual were analyzed by NanoDrop 1000 (Thermo Scientific) and BioAnalyzer II (Agilent Technologies). The RNA samples from the same litter that passed the quality control (RNA integrity number [RIN] > 8.0, A260/280, and A260/230 > 1.9) were used for sequencing library constructions. Paired-end 2 × 101 nt sequencing for juvenile samples and paired-end 2 × 150 nt sequencing for embryonic samples were conducted by the High Throughput Genomics Core Facility, Biodiversity Research Center, Academia Sinica, Taiwan, and Novogen, United States, respectively. The raw QC data are shown in Source Data file.

### Manual annotation of keratin genes
The annotations of the latest chicken genome assembly (GG6a) were downloaded from Ensembl[80] and RefSeq[81]. The Hidden Markov Models (HMMs) that represent alpha-keratins and beta-keratins were

downloaded from Pfam 33.1[82]. The putative alpha-keratins and beta-keratins in Ensembl and RefSeq annotations was predicted using the Hmmsearch function in HMMER 3.1 (https://hmmer.org)[83]. For alpha-keratins, we further eliminated the genes that were not located in the typical type I and type II alpha-keratin gene clusters. Finally, the predicted alpha-keratin genes and beta-keratin genes in Ensembl and RefSeq annotations were further compared to the annotation in ref. 84, and the final annotation of alpha-keratins and beta-keratins was obtained by manual judgement for the most appropriate gene models. The GTF file of the annotated keratin genes is shown in the Source Data file.

### RNA-seq analysis

For each transcriptome, low-quality reads were trimmed and adapters were removed using Trimmomatic[85]. The processed paired-end reads were mapped to the reference genomes of chicken (GRCg7w for TO-GCN and manually-annotated GRCg6a for keratin analysis) using Hisat2 with default settings[86]. Two genomes were used because the manually annotated GRCg6a was made before GRCg7w was released. From the aligned reads the transcripts were assembled using StringTie with annotation-based settings (with −e)[87]. The gene expression level was calculated in terms of Transcripts Per Kilobase Million (TPM). For TPM calculations, uniquely mapped reads were first assigned to the genes. Multiple-hit reads were then redistributed to genes based on their relative abundances of uniquely mapped reads. A gene is considered expressed if its TPM is $\geq 1$ in at least one of the transcriptomes. To compare expression profiles of the selected genes across the conditions, we applied the upper quartile normalization procedure[88]. Differential expression analysis between different samples was calculated using the R package DESeq2[89].

### Construction of TO-GCNs

Our comparative transcriptomics method[25] was designed to analyze time-course or developmental-stage transcriptomes that have five or more different conditions between the two feather types: E7 epidermis, E7 dermis, E9 epidermis, E9 dermis, and E14 feather filaments in posterior dorsal regions for embryonic samples; D3, D4, D5, D6, and D7 posterior dorsal skins for juvenile samples. The method consists of three steps: determining co-expression cutoffs, constructing GCNs, and determining the time order of TF gene expression to transform a GCN into a TO-GCN. First, the Pearson correlation coefficient (PCC) values of all TF–TF gene pairs were calculated and used to determine the cutoffs of co-expression. Second, using the co-expression relationships between TF genes, we determined the GCN for connected TF genes. Third, the time order of TF genes in the GCN was assigned by the breadth-first search (BFS) algorithm[90] initiated from the selected node that should be the first upregulated TF in the GCN. BFS is an algorithm for searching a network graph. It starts with an initial seed and searches all its neighbors (nodes with connecting edges) to form a set of nodes (level 1). Then, the process proceeds from all nodes in level 1 and searches their neighbors (excluding level 1 nodes) to form the second set of nodes (level 2) and so on, until all nodes in the network are assigned. The computer programs for the method are available at https://github.com/petitmingchang/TO-GCN (31). SP5 and MYOD1 TF genes were selected as the initial nodes in E7 embryonic samples and D3 juvenile samples, respectively.

### Co-expressed gene sets and overrepresented functions at each TO-GCN level

For the TF genes at each level of a TO-GCN, a corresponding set of co-expressed genes (usually non-TF genes) can be identified with the same co-expression relationship for adding the genes to the TO-GCN. Since a gene may be co-expressed with TF genes in multiple levels, two neighboring gene sets may have some overlapping genes. For each set of genes corresponding to a level in a TO-GCN, a functional enrichment

analysis was conducted with the background set of all expressed genes in this study. Fisher's exact test with the false discovery rate (FDR) < 0.05 was applied with functional annotations from Reactome (https://reactome.org/).

### Statistical analysis

To determine the cutoff of PCC (Pearson correlation coefficient) values between the expression profiles of two TF genes for building a TO-GCN, the PCC values of gene expression profiles from all TF-TF gene pairs were calculated. For each time course, the PCC values were collected to generate an empirical distribution of PCC values for each set of time-course transcriptomic data[25]. Then the cutoff was determined as 0.9 (embryo) and 0.92 (juvenile) based on the $p$-value < 0.05 from the right end of the empirical distribution for each TO-GCN[25]. In comparing two TO-GCNs, all differences in levels (embryonic–juvenile) were collected to generate a histogram (empirical distribution) for classifying TFs into two categories of changed (difference $\geq 3$) and not changed (difference $\leq 2$) levels based on 1.5x standard deviation. In pathway enrichment analysis, only pathways with FDR < 0.05 were selected. In DEG analysis, the differentially expressed genes were defined by the adjusted $p$-value < 0.05 and the absolute $\log_2$ fold change > 1 (Supplementary Data 7).

### ATAC-seq libraries construction

The soft tissues of adult flight feather follicles were dissected and dissociated. Specifically, adult tissues were dissociated in 100-μl of dissociation solution containing 0.1% collagenase Type I and 0.25% Trypsin in 1x PBS and incubated in 37 °C MultiTherm Shaker (Southern Labware) for 15 min with shaking at 1500 rpm. The ATAC-seq libraries were constructed as follows: 3000 cells were pelleted and resuspended in 20 μl lysis buffer containing 10 mM Tris HCl (pH = 8.0), 5 mM MgCl$_2$, 10% DMF, and 0.2% NP-40 (11332473001, Roche). Lysis was performed on ice for 15 min before adding 30 μl reaction buffer containing 10 mM Tris HCl (pH = 8.0), 5 mM MgCl$_2$, 10% DMF, and 0.5 μl Tn5 Transposase (Tagment DNA Enzyme I, 15027865, Illumina). The transposition reactions were performed at 37 °C for 20 min. DNA was column purified (D4014, Zymo), and the transposed DNA was amplified with a modified version of custom-designed primers[29]. The reaction was monitored after five cycles with qPCR using a 1/10 transposition reaction mixture and the corresponding primers in a total volume of 15 μl (KK4617, Sigma). Cycles were added as calculated, and then the amplified samples were purified and size selected using AMPure beads (A63881, Beckman Coulter). Libraries were quality controlled using the TapeStation (G2992AA, Agilent), and 150 bp paired-end sequencing was performed on the NovaSeq 6000 (Novogen, United States).

### ATAC-seq basic analysis

Cutadapt was used to remove adapter sequences with the following parameters:−q 20−cut 1−length 75−minimum-length 36[91]. Hisat2 with the following parameters was applied to reads mapping:−no-temp-splice-site−no-spliced-alignment−X 2000[86]. Samtools was used to remove low-quality and non-unique hits with the parameters: view -b -f 3 -F 4 -F 8 -F 256 -F 2048 -q 30[92]. After removing the mitochondria alignments, the Picard toolkit was used to remove the PCR duplicates (http://broadinstitute.github.io/picard/). MACS2 was used to call peaks with the following parameters:−nomodel−shift -37−extsize 73−keep-dup all−SPMR -f BAMPE[93].

### Footprinting analysis of ATAC-seq data

To identify, plot, and compare transcription factor footprints in the different samples, we used the program RGT HINT-ATAC[94], a hidden Markov model (HMM) based predictor. In detail, the BAM files containing the filtered aligned reads for each biological replicate were merged and used as the matrix for the footprinting analysis for the regions corresponding to the peaks called by MACS2 (the narrowpeak

files). The differential TF bindings between embryonic and adult feather follicles were compared based on JASPAR version 2020 with default settings[94,95]. Embryonic ATAC-seq data were from the previous study[27]. The raw data are shown in the Source Data file. The ATAC peaks were visualized by Gviz[96].

## Section in situ hybridization (SISH)
To generate RNA probes, PCR was performed using cDNA from E14 feather filaments or adult flight feathers. The following prime pairs were used: *CTNNB1* forward: GCAACTCGTGCAATCCCAGA; reverse: CAAAGGCCAGTGTGAGGGTG, *LEF1* forward: TCAAGTCCTCGCTGGTC AAC; reverse: GGACATGGAAGGGTCGACTG, *GG6AChr25Scale2* forward: CAGTGCCCCGACTCAACG; reverse: AGCAACTAAGAAGA-CAGGGACT, *GG6AChr25Scale10* forward: ATGTCTTGCTCCGACCTGT; reverse: GGGATGTGAAGCTGATAGCATTG, *BMP4* forward: TTCCAC-CATGAAGAGCACCTG; reverse: CAACCCACGTCGCTGAAATC, *SOX14* forward: CTCCTTACTTGACCCCAGCCA; reverse: CGACCAAGCGGTA-CAGTTACAC, *TWIST2* forward: TTTCCAAAGGATCTGTCTCAGGA; reverse: TAGTGCGAGGCTGACATGGA, *SHH* forward: CTGGTGAAG-GACCTGAGCCCT; reverse: GCCCAACTGTGCTCCTCGAT. The two scale probes can hit around three scale keratin genes, but two hit groups were not overlapped. For in situ hybridization, sections were cleared of paraffin wax by incubating twice in xylene for 10 min each. Rehydration was performed through an ethanol series. Sections were washed twice with PBS for 5 min each. Proteinase K treatment (5 mg/ml) was performed for 10 min to enhance permeability. Sections were rinsed again with PBS twice (5 min each). Post-fixation was performed with 4% paraformaldehyde for 20 min. After PBS washes, sections were incubated in 0.1 M triethanolamine buffer (pH 8.0) for 10 min and acetylation was performed using 0.25% acetic anhydride in triethanolamine buffer for 10 min. Sections were then washed with 2X SSC solution. Dehydration was performed again through an ethanol series, followed by air-drying for 30 min. Hybridization solution containing formamide, SSC, blocking reagent, and probe (100 ng/ml) was applied to each slide. Slides were incubated with coverslips at 65 °C for more than 8 h. Stringent washes were performed with decreasing SSC concentrations (2X SSC and 0.2X SSC) at 65 °C for 20 min each (3 times each). After blocking with 20% goat serum for 2 h, anti-digoxigenin antibody (diluted 1:1000, 11093274910, Sigma) was added to the slides for incubating at 4 °C overnight. Extensive washing steps were performed with PBS (4 times, 30 min each) and NTMT (0.1 M Tris-HCl, pH 9.5, 50 mM MgCl₂, 0.1 M NaCl, 0.1% Tween 20) buffer (2 times, 15 min each). Color development was achieved using NBT/BCIP (S3771, Promega) chromogenic substrates.

## Construction and misexpression of dominant negative form of the candidate genes
To perform tissue specific gene knock down, the misexpression of sequence modified genes were used to achieve the dominant negative (dn) effect. The RCAS-dnLEF1 has been applied in chicken studies and was obtained from Addgene (Plasmid #14022)[97]. The dnACTA2 function was published previously[48], and the sequence was cloned into RCAS-cherry plasmid in this study. Virus was made according to ref. [78] and concentrated by ultracentrifugation. The flight feathers of SPF chickens were plucked and around 50ul of concentrated RCAS virus was injected into each follicle cavity. All the experiments were conducted in both sexes of chickens to eliminate the probable sexual specific effects. The flight feathers that were plucked at the same time without virus injection were used as the control. Feather follicles including dermal papilla were dissected after two weeks of the virus injection and fixed immediately in 4% PFA.

## Reporting summary
Further information on research design is available in the Nature Portfolio Reporting Summary linked to this article.

## Data availability
The authors declare that all data supporting the findings of this study are available within the article and its Supplementary Information files, or are available from the authors upon request. The source data for Figs. 2, 4, and 5, Methods, Supplementary Figs. 3–5, and Table 1 are provided as a Source Data file. High throughput sequencing data have been deposited in the NCBI Sequence Read Archive (SRA) under Bio-Project ID: PRJNA1084783. Source data are provided with this paper.

## Code availability
The codes used for TO-GCN analyses are available at https://github.com/petitmingchang/TO-GCN. The codes used for ATAC peak visualization and other analysis are available at https://github.com/r93b42016/feather_transition.

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

## Acknowledgements

We thank Prof. Gunter Wagner for his helpful suggestions. We thank Nicole E. Schweers, California Institute of Technology, for offering us extra samples for paper revision. We thank Gee-Way Lin, Taipei Medical University, for providing zebra finch scheme templates. We thank the National Core Facility for Biopharmaceuticals (NCFB, 112-2740-B-492-001) and the National Center for High-performance Computing (NCHC) of National Applied Research Laboratories (NARLabs) of Taiwan for providing computational resources and storage resources. We also thank the Center for Advanced Research Computing (CARC, URL: https://carc.usc.edu.) at the University of Southern California for pro-viding computing resources. RCAS(A) mdnLef1en was a gift from Cliff Tabin (Addgene plasmid # 14022). The Taiwan team was supported by the National Science and Technology Council, Taiwan (NSTC 112-2311-B-045-3). The USC team was supported by NIH grants RO1 AR 047364, R37 AR 060306, and the China Medical University (in Taiwan)—University of Southern California collaborative grant 005884-00001. C.K.C. is sup-ported by the iEGG and Animal Biotechnology Center from the Feature Areas Research Center Program within the framework of the Higher Education Sprout Project by the Ministry of Education (MOE-112-S-0023-A) in Taiwan. T.Y.L. is supported by the Dragon Gate Program (112-2926-I-006-507-G) in Taiwan. Z.Y. is supported by the USC CIRM COMPASS Scholars Program (EDUC4-12756) in the USA.

## Author contributions

Conceptualization: C.-K.C., W.-H.L., C.-M.C.; Methodology: C.-K.C., Y.-M.C., T.-X.J., P.W.; Histology: C.-K.C., T.-X.J., T.-Y. L., T.-D.V., T.-Y.H., P.W., H.I-C.H.; Bioinformatic analysis: C.-K.C., Y.-M.C., J.-J.L.; Experimental validations: C.-K.C., T.-X.J., Z.C.Y., P.W., T.-Y.L., J.L., Z.Y., H.I-C.H.; Resources: C.-K.C. and C.S.N.; Writing and editing: C.-K.C., Y.-M.C., P.W., W.-H.L., C.-M.C.; Supervision and funding acquisition: W.-H.L., C.-M.C.

## Competing interests

The authors declare no competing interests.
