## [Peer Review File · Nature Communications]

Transition from natal downs to juvenile feathers: conserved regulatory switches in birdsReviewer #1 (Remarks to the Author):

In this article, Chen et al. study the molecular events occurring during primary feather transition (i.e., the replacement, in the same feather follicle, of radially-symmetric natal down by bilaterally-symmetric juvenile flight feathers, a robust phenomenon occurring in all avian species regardless of their life history).

Using thorough histological analyses in chicken and zebra finch embryonic and juvenile specimens, authors define 5 major morphological changes characterizing primary feather transition, namely (successively) biconcave dermal papilla formation, peripheral pulp formation, rachis formation, feather sheath thickening and vane formation. They then perform transcriptomics analyses at each stage, and identify a number of differentially regulated pathways, which they test in vivo using classical expression analyses (in situ hybridization) and technically challenging functional analyses (retrovirus-mediated expression of dominant negative forms in cavities of plucked juvenile feathers or over-expression in forming follicles).

They show that (1) the Wnt effector LEF1, expressed in an antero-posterior gradient during rachis formation, is necessary for rachis production, (2) the extra-cellular matrix likely controls peripheral pulp formation, consistent with previous results in adult chicken feathers, and (3) α -SMA is expressed in juvenile dermal papillae, where it is required for feather follicle differentiation. Finally, by combining further analysis of transcriptomics data with ATAC sequencing, they show that the expression of scale keratin genes is specific to juvenile feathers, and is controlled by the transcription factor Sox14.

This is an impressive study with regards not only to the significance of the results that will be of interest to a large community studying feather production and evolution, but also to the amount of genomics data gathered, and expression and functional tests performed. It is certainly worth publication in Nature Communications, provided the authors can address the following concerns:

- The manuscript is unevenly written: the introduction and discussion are written in a clear, readable fashion while the result section, which is denser –perhaps because it describes a large amount of data, sometimes reads as the juxtaposition of separate datasets rather than the unfolding of a “story”. The manuscript would benefit from re-writing in many places to air and better structure the different paragraphs. The title of each paragraph often represents an over-interpretation of results (see below) or does not clearly relate to results. It could also help to reorder results so as to present the molecular analyses in the same order as the timely transition of feather types (i.e., with results on dermal papillae first, followed by results on rachis formation, etc).

- Results from expression analyses are often difficult to interpret because of low magnification of pictures or of the weakness of stains; this reviewer is not convinced by several observations of differences in spatial expression or level of expression of candidate factors analyzed, and strongly suggests complementing qualitative assessments with qualitative experiments (qPCRs).

- Figure 1: The schematics in (h) are not clear: do they represent timely dynamics? What is the relationship between the first and the second schematics? Please provide additional explanation. It would also be more consistent to show a schematics representing the zebra finch juvenile. Why is the timely analysis not performed in the zebra finch as well, contrary to all other parts of the study?

Line 111: the term “evolutionary morphogenesis events” is an overstatement; there is no evolutionary evidence or description of morphogenetic processes in the paper.

- Figure 2: This reviewer is not certain that the TF-specific analysis is relevant as performed here (TF specificity is decided on the basis of a difference between embryonic and juvenile expression above/under a given threshold). How is this threshold chosen? Do conclusions hold when its value is changed? In my view, this part could be removed or should be better justified.

There is no literature reference for the function of PRC2.

Fig 2d panel is too small for proper data appreciation from the reader.

- Figure 3: Functional tests are elegant and convincingly support the conclusions, but expression

analyses are not clear: from the provided image, one cannot clearly see that LEF1 is expressed in an AP gradient.

What is CTNNB1, and why is it used? Please explain.

Line 171: "the molecular gradient": what does it refer to?

Line: 175: Please revise the syntax of the sentence: "could be redundant or regional specificity"

Line 181: There is no literature reference for the function of LEF1

- Figure 4: The effects of dominant negative α -SMA expression are too severe to conclude that this molecule plays a specific role in primary feather transition. A novel technical set up to obtain a less drastic, interpretable phenotype would be understandably difficult to establish, but authors should nonetheless use more careful phrasing in their conclusions: as such, the hypothesis that " α -SMA not only is a structural protein but also establishes an environment for the primary feather transition" does not appear strongly supported.

What does the term "cyclic renewal" in the title of the paragraph refers to, with regards to the results described?

- Figure 5: This part of the paper is truly admirable for the combination of techniques used; it makes a convincing case for a role of Sox14 in the control of keratin production in the feather sheath. It would be nice if better / higher magnification pictures could be provided for expression analyses: at the present magnification, it is difficult to see the increase in feather sheath thickness (or the red arrows) in the images. It is also difficult to observe changes in expression levels for Sox14 and other markers, and as mentioned above, for data robustness, this reviewer would recommend performing quantitative experiments.

- Discussion: the discussion spans various fascinating topics based on results obtained in this study, from the role of Wnt factors or the ECM in different events characterizing primary feather transition to the similitude observed in these processes in two different bird groups represented by the chicken and the zebra finch. It would be interesting to also discuss the link between molecular events implicated here and changes in the symmetry of the feather during primary transition (from radial to bilateral).

In the discussion and several places in the paper (e.g., lines 83-86 / 336 / 375), authors imply that chicken are part of the Neoaves group, but if I am not mistaken these birds are part of the Galloanserae group. This must be corrected (if anything, the fact that the chicken is part of a different group strengthens the author's conclusions on the conservation of processes of primary feather transition).

Line 331: "to strengthen the sheath of juvenile feathers": this is not shown in this study.

Reviewer #2 (Remarks to the Author):

In this manuscript, four villus-juvenile feather transition events were elucidated by time-ordered gene co-expression network construction, epigenetic analysis, and functional perturbations in developing feather follicles, The roles of LEF1, ACTA2 and SOX14 in these processes were also revealed. This study adds valuable information for understanding Evo-Dove. Below are my minor comments:

1. Please use the latest or representative research reports, and optionally delete some references.
2. Line 70-86: Please rewrite this section, it was a little long.
3. Please use the " μ m" instead of "um" in the figure comments.
4. Please indicate "P" and "R" in Supplementary Figure 3b.
5. Please rewrite the statistical analysis method separately and clearly in the section of "Materials and methods".

Reviewer #3 (Remarks to the Author):

In the paper entitled «Transition from natal downs to juvenile feathers : conserved regulatory switches in Neoaves », the authors, Chen et al., investigate the formation, mainly at the descriptive level and identification of molecular actors, of early feathers in two avian species, the chicken and the zebrafinch, representative of two avian phyla.

As a general comment, the manuscript is very clear, very complete and highly remarkable for the amount of data and new characterizations in EDC genes, particularly in the distribution of the β keratins. Only a few points need to be clarified :

- i) the authors mention - muscle bundles'- 'square... to diamond'..., notably in figure 1b'-d', but it would be clearer to indicate them, materialize them (lines?) because the contours are not obvious. On what basis other than morphological are these structures defined?
- ii) in the TO-GCN method, 11 clusters are identified but with very different gene levels between the groups. It is mentioned that there are no differences - similarities - between the embryonic and juvenile stages. This is a little surprising because the kinetics are not the same with gene 'peaks' at different levels. The number of genes is also sometimes very significantly different. Could the authors comment and modify the text accordingly?
- iii) It is not clear whether the same TF follows the same profile between the two stages and what are the genes on which the differences are based? It would be interesting to illustrate the profiles with some of the most differentially expressed TFs depending on the stages.
- iv) In the paragraph 'in the juvenile TO-GCN, it would also be advantageous to mention some of the most expressed genes in order to better illustrate the text.
- v) the choice of level $d > \text{or} < \pm 3$ is not clearly explained compared to ± 2 ?
- vi) In Figure 3, it would be interesting to mention the frequencies observed in the alterations obtained with RCAS-dnLEF1. Is the penetrance of the phenotype complete - 100% and how is the θ angle distributed?
- vii) For the SOX14 experiments, the authors mention SOX14 as a major player in the differentiation of the spinal cord... it should just be clarified that the stage of development is not at all the same as that proposed for the development of feather sheath, even if this seems obvious. more importantly, the co-regulations between the TFs SOX14, ESRRB, ESRRG, PRDM4, SREBF2 and SMAD5 are also not clearly identified - knowing that some of these genes are also involved in other processes - including cell renewal. stem cells for example, - and that relative expression levels may not represent the dynamics and importance of these genes. Would certain ISH demonstrate expression in some stem cells ensuring the production of these feathers? Could the authors comment and hypothesize this in the text?

Minor typing errors :

- Juvenivel in fig 2 (not juvenile)
- Fig 2a-2e (not a2-e2)

In conclusion, the manuscript is remarkable, very complete - sometimes very complex due to the quantity of data provided - but deserves full attention for publication once the answers have been provided to the few points raised

REVIEWER COMMENTS

Reviewer #1 (Remarks to the Author):

In this article, Chen et al. study the molecular events occurring during primary feather transition (i.e., the replacement, in the same feather follicle, of radially-symmetric natal down by bilaterally-symmetric juvenile flight feathers, a robust phenomenon occurring in all avian species regardless of their life history).

Using thorough histological analyses in chicken and zebra finch embryonic and juvenile specimens, authors define 5 major morphological changes characterizing primary feather transition, namely (successively) biconcave dermal papilla formation, peripheral pulp formation, rachis formation, feather sheath thickening and vane formation. They then perform transcriptomics analyses at each stage, and identify a number of differentially regulated pathways, which they test in vivo using classical expression analyses (in situ hybridization) and technically challenging functional analyses (retrovirus-mediated expression of dominant negative forms in cavities of plucked juvenile feathers or over-expression in forming follicles).

They show that (1) the Wnt effector LEF1, expressed in an antero-posterior gradient during rachis formation, is necessary for rachis production, (2) the extra-cellular matrix likely controls peripheral pulp formation, consistent with previous results in adult chicken feathers, and (3) α -SMA is expressed in juvenile dermal papillae, where it is required for feather follicle differentiation. Finally, by combining further analysis of transcriptomics data with ATAC sequencing, they show that the expression of scale keratin genes is specific to juvenile feathers, and is controlled by the transcription factor Sox14.

This is an impressive study with regards not only to the significance of the results that will be of interest to a large community studying feather production and evolution, but also to the amount of genomics data gathered, and expression and functional tests performed. It is certainly worth publication in Nature Communications, provided the authors can address the following concerns:

- The manuscript is unevenly written: the introduction and discussion are written in a clear, readable fashion while the result section, which is denser –perhaps because it describes a large amount of data, sometimes reads as the juxtaposition of separate datasets rather than the unfolding of a “story”. The manuscript would benefit from re-writing in many places to air and better structure the different paragraphs. The title of each paragraph often represents an over-interpretation of results (see below) or does not clearly relate to results. It could also help to reorder results so as to present the molecular analyses in the same order as the timely transition of feather types (i.e., with results on dermal papillae first, followed by results on rachis formation, etc).

Thanks for the suggestion, we have re-ordered the results.

- Results from expression analyses are often difficult to interpret because of low magnification of pictures or of the weakness of stains; this reviewer is not convinced by several observations of differences in spatial expression or level of expression of candidate factors analyzed, and strongly suggests complementing qualitative assessments with qualitative experiments (qPCRs).

Thanks for the suggestions. QPCR is a great way to quantify and compare the gene expressions in specific tissues. However, feather follicles are extremely difficult to dissect precisely and biased micro-

dissections can make the results biased a lot. For example, ACTA2 is expressed in dermal papilla; however, it is also strongly expressed in the beneath muscle tissues. If the whole follicles were dissected, the remnant muscle tissues will contribute to the expression values of ACTA2, leading to the loss of significance between feather follicle comparisons. Moreover, it is very difficult to know whether the dissection is good enough. This is also why most feather/hair follicle studies only conducted *in situ* hybridization or IHC to detect the gene expression patterns. We therefore applied two ways to address this issue. First, we performed triplicates for the post-hatch samples to achieve reasonable statistics. Second, instead of simply using gene differential expression analysis, we conducted TO-GCN analysis to identify the candidate pathways because the time course analysis can overcome the tissue complexity issue (which always generates too many candidates in pairwise comparisons). Therefore, to make our results more convincing, we updated most of our staining and added more finch data for the comparisons (Fig. 1).

- Figure 1: The schematics in (h) are not clear: do they represent timely dynamics? What is the relationship between the first and the second schematics? Please provide additional explanation.

Thanks for clarifying what we want to show to the readers. Fig. 1h has now been moved to Fig. 1k. We also added several descriptions in the first result section to explain our findings using the schemes in Fig. 1.

It would also be more consistent to show a schematic representing the zebra finch juvenile. Why is the timely analysis not performed in the zebra finch as well, contrary to all other parts of the study?

Thanks for raising the question. Zebra finch samples are actually not easy for us to get, especially in this non-breeding season (we only have outdoor aviary). We tried our best to fill up the gaps and now the comparison should be more complete (Fig. 1). Unfortunately, we are not able to make good longitudinal sections of early juvenile follicles in zebra finches because: 1. the follicles do not grow in regular array like those in chickens, instead, they grow radially, 2. the follicle number is low, and 3. they are not visible yet like those in D7 zebra finches (see Fig. 1b-d).

Line 111: the term “evolutionary morphogenesis events” is an overstatement; there is no evolutionary evidence or description of morphogenetic processes in the paper.

Thanks for the correction. We have replaced “evolutionary morphogenesis events” by “morphogenesis events”.

- Figure 2: This reviewer is not certain that the TF-specific analysis is relevant as performed here (TF specificity is decided on the basis of a difference between embryonic and juvenile expression above/under a given threshold). How is this threshold chosen? Do conclusions hold when its value is changed? In my view, this part could be removed or should be better justified.

Thanks for the comments. We wrote a new section, Statistical Analysis, in the Methods to clarify the cut-offs we set to construct the gene regulatory networks. Moreover, we no longer rely on level differences since it is indeed artificial. Instead, we listed all the possibilities (Fig. 2c) and validated the genes with literature or independent analysis supports. Finally, as mentioned above, TO-GCN is a necessary way to decipher gene regulatory differences in complex tissues, so we kept it.

There is no literature reference for the function of PRC2.

Thanks for notifying it. A reference is added.

Fig 2d panel is too small for proper data appreciation from the reader.

Thanks for the suggestion. Fig 2d has been removed since it is not essential. We didn't describe the results of the heatmap. Instead, we highlighted most possible candidates in Fig. 2c and validated two of them in Fig. 2d. The reasons were explained in the last paragraph of "Coordination of multiple signaling pathways during the primary feather transition".

- Figure 3: Functional tests are elegant and convincingly support the conclusions, but expression analyses are not clear: from the provided image, one cannot clearly see that LEF1 is expressed in an AP gradient.

Thanks for the comment. We replaced it by a picture showing better staining. Also, we replaced the overall skin view with embryonic feather follicles to show the expression difference between the two feather types (Fig. 4b).

What is CTNNB1, and why is it used? Please explain.

Thanks for the question. CTNNB1 (beta-catenin) is usually the co-factor of LEF1 and important for embryonic feather initiation. Our finding suggests that CTNNB1 may not be the key factor for rachis formation. We have added the description in the second paragraph of "Wnt gradient is the major regulator of rachis formation and LEF1 is a key molecular hub converting radial downy to bilaterally symmetric juvenile feathers".

Line 171: "the molecular gradient": what does it refer to?

Please kindly see the renewed Fig. 4b.

Line: 175: Please revise the syntax of the sentence: "could be redundant or regional specificity"

Thanks for the correction. "could be redundant or regional specificity" is replaced by "could be redundant or region specific".

Line 181: There is no literature reference for the function of LEF1

Thanks. References are added.

- Figure 4: The effects of dominant negative α -SMA expression are too severe to conclude that this molecule plays a specific role in primary feather transition. A novel technical set up to obtain a less drastic, interpretable phenotype would be understandably difficult to establish, but authors should nonetheless use more careful phrasing in their conclusions: as such, the hypothesis that " α -SMA not only is a structural protein but also establishes an environment for the primary feather transition" does not appear strongly supported.

Thanks for the suggestion. Indeed, we overstated the function of ACTA2. " α -SMA not only is a structural protein but also establishes an environment for the primary feather transition" is rephrased to " α -SMA could serve as a structural component to build the micro-environment for the primary feather transition in birds".

What does the term "cyclic renewal" in the title of the paragraph refers to, with regards to the results described?

Thanks for pointing out our insufficient description. We made two changes: 1. In introduction, “In adult chickens, feathers need to undergo molting to maintain their normal functions and the molting process is known to be controlled by the dermal papilla and regulated by Wnt inhibitors” was changed to “In adult chickens, feathers need to undergo molting to maintain their normal functions and the molting process, as known as cyclic renewal, is controlled by the dermal papilla and regulated by Wnt inhibitors”. 2. In the paragraph of cyclic renewal, “Dermal papilla located at the follicle base is essential for cyclic renewal. In our observation, juvenile and adult feathers undergo cyclic renewal but natal downs do not have the regeneration ability. Here we mimic the feather cyclic renewal by juvenile feather plucking and regeneration” was added to the leading sentence.

- Figure 5: This part of the paper is truly admirable for the combination of techniques used; it makes a convincing case for a role of Sox14 in the control of keratin production in the feather sheath. It would be nice if better / higher magnification pictures could be provided for expression analyses: at the present magnification, it is difficult to see the increase in feather sheath thickness (or the red arrows) in the images. It is also difficult to observe changes in expression levels for Sox14 and other markers, and as mentioned above, for data robustness, this reviewer would recommend performing quantitative experiments.

Thanks for the suggestions. Since the feather sheath is only a small portion of a follicle, we added Supplementary Figure 5b to increase the magnification of Fig. 5l. Also, we compared the thicknesses of feather sheath between the virus-infected and control follicles. The two feather follicles are adjacent in the skin with the same magnification, so the consequence of SOX14 expression on the dermal sheath formation can be compared in the enlarged figure.

- Discussion: the discussion spans various fascinating topics based on results obtained in this study, from the role of Wnt factors or the ECM in different events characterizing primary feather transition to the similitude observed in these processes in two different bird groups represented by the chicken and the zebra finch. It would be interesting to also discuss the link between molecular events implicated here and changes in the symmetry of the feather during primary transition (from radial to bilateral).

Thanks for the great suggestion. We added Supplementary Figure 4g and a new paragraph in the Discussion: The five molecular events are basically independent because they appear in different developmental time frames. How the sequential morphogenesis is established is an interesting question. Whether the morphogenesis interacts with each other is another interesting issue that remains to be investigated further. It seems that the establishment of organized dermal papilla is the most basic step because the overexpression of dnACTA2 not only stops the feather regeneration but also disable the barb formation (Supplementary Figure 6), thus the rachis and barbule hooklets failed to form. The only remaining phenotypic structure is the feather sheath, suggesting that feather sheath formation is an independent event. Moreover, IHC of β -keratin in dnLEF1 overexpressed follicle shows similar pattern with that in control follicle (Fig. 4), suggesting that rachis formation and barbule hooklet formation are two independent events.

In the discussion and several places in the paper (e.g., lines 83-86 / 336 / 375), authors imply that chicken are part of the Neoaves group, but if I am not mistaken these birds are part of the Galloanserae group. This must be corrected (if anything, the fact that the chicken is part of a different group strengthens the author's conclusions on the conservation of processes of primary feather transition).

Thanks a lot for pointing this out. We found this error after the paper submission. Now we replaced “Neoaves” by “birds”, and mentioned that barbule hooklets have been lost in most ratites.

Line 331: “to strengthen the sheath of juvenile feathers”: this is not shown in this study.

Please kindly see Supplementary Figure 5b.

Reviewer #2 (Remarks to the Author):

In this manuscript, four villus-juvenile feather transition events were elucidated by time-ordered gene co-expression network construction, epigenetic analysis, and functional perturbations in developing feather follicles. The roles of LEF1, ACTA2 and SOX14 in these processes were also revealed. This study adds valuable information for understanding Evo-Dove. Below are my minor comments:

1. Please use the latest or representative research reports, and optionally delete some references.

Thanks for the suggestion. We have removed the old or non-representative papers from the sentences with more than 5 references to make the article more concise.

2. Line 70-86: Please rewrite this section, it was a little long.

Thanks for the suggestion. We have re-written it.

3. Please use the “ μm ” instead of “um” in the figure comments.

Thanks for the correction. We have corrected them.

4. Please indicate “P” and “R” in Supplementary Figure 3b.

It was actually an error. Thanks for pointing it out and we have updated the figure.

5. Please rewrite the statistical analysis method separately and clearly in the section of “Materials and methods”.

Thanks for the suggestion. We have added “Statistical Analysis” to the Methods.

Reviewer #3 (Remarks to the Author):

In the paper entitled «Transition from natal downs to juvenile feathers: conserved regulatory switches in Neoaves », the authors, Chen et al., investigate the formation, mainly at the descriptive level and identification of molecular actors, of early feathers in two avian species, the chicken and the zebra finch, representative of two avian phyla.

As a general comment, the manuscript is very clear, very complete and highly remarkable for the amount of data and new characterizations in EDC genes, particularly in the distribution of the beta keratins. Only a few points need to be clarified:

i) the authors mention - muscle bundles'- 'square... to diamond'...., notably in figure 1b'-d', but it would be clearer to indicate them, materialize them (lines?) because the contours are not obvious. On what basis other than morphological are these structures defined?

Thanks for the good question. We have thought it over and now believe that the muscle bundle changes could be from the stretching of the skins. The embryonic skins are easy to be fixed but the fixations of the juvenile skin require extra forces. The extra forces from our hands could be the main reason for the muscle bundle angle changes. We have removed this sentence.

ii) in the TO-GCN method, 11 clusters are identified but with very different gene levels between the groups. It is mentioned that there are no differences - similarities - between the embryonic and juvenile stages. This is a little surprising because the kinetics are not the same with gene 'peaks' at different levels. The number of genes is also sometimes very significantly different. Could the authors comment and modify the text accordingly?

Thanks for the comment. Indeed, the TO-GCN profiles between the two feather types are not really similar. We removed this overstatement.

iii) It is not clear whether the same TF follows the same profile between the two stages and what are the genes on which the differences are based? It would be interesting to illustrate the profiles with some of the most differentially expressed TFs depending on the stages.

Thanks for the great suggestion. In response, we have highlighted several genes from the keratinization stages (levels 8 to 10, please see the last paragraph of “Coordination of multiple signaling pathways during the primary feather transition”). We picked up ID3, which is expressed in the same level of both species, and BNC1, which is expressed distinctly between the two feather types, and verified them by in situ hybridizations to prove that level difference can represent phenotypic difference at least at the keratinization stage (Fig. 2d).

iv) In the paragraph 'in the juvenile TO-GCN, it would also be advantageous to mention some of the most expressed genes in order to better illustrate the text.

Thanks for the suggestion. We have highlighted them as mentioned above.

v) the choice of level $d >$ or $< +/- 3$ is not clearly explained compared to $+/- 2$?

Thanks for pointing this out. Yes, we do think the criteria of $+/- 3$ is too artificial and even 1 level difference can reflect totally different regulations. Instead of setting this cut-off, we now highlighted all

the possible candidates and functional validated them based on literatures or independent analysis (Fig. 2c).

vi) In Figure 3, it would be interesting to mention the frequencies observed in the alterations obtained with RCAS-dnLEF1. Is the penetrance of the phenotype complete - 100% and how is the \square angle distributed?

This is an interesting question. However, with the current RCAS technique in the chicken, the virus infection sites and dosages are hard to control and be precisely quantified. We did many experiments (n= 30) and they usually showed reduction of the theta angle (20 out of 30). Also, we noted the more reduction of the theta angle, the smaller the rachis.

We have added additional representative phenotypic changes in Supplementary Figure 4h.

vii) For the SOX14 experiments, the authors mention SOX14 as a major player in the differentiation of the spinal cord... it should just be clarified that the stage of development is not at all the same as that proposed for the development of feather sheath, even if this seems obvious. more importantly, the co-regulations between the TFs SOX14, ESRRB, ESRRG, PRDM4, SREBF2 and SMAD5 are also not clearly identified - knowing that some of these genes are also involved in other processes - including cell renewal. stem cells for example, - and that relative expression levels may not represent the dynamics and importance of these genes. Would certain ISH demonstrate expression in some stem cells ensuring the production of these feathers? Could the authors comment and hypothesize this in the text?

Thanks for raising this issue. We removed the sentence: "Related studies are limited but SOX14 is essential for the initiation of neuronal differentiation in the chick spinal cord". The discussion of SOX14, ESRRB, ESRRG, PRDM4, SREBF2 and SMAD5 were added to the 4th paragraph of "Many scale keratins are specifically upregulated in juvenile feather sheath by SOX14" and SISH of SREBF2 were added to Supplementary Figure 5b.

Minor typing errors :

- Juvenivel in fig 2 (not juvenile)

Thanks for the correction. Corrected.

- Fig 2a-2e (not a2-e2)

They actually mean Fig 4a2 and Fig 4e2.

In conclusion, the manuscript is remarkable, very complete - sometimes very complex due to the quantity of data provided - but deserves full attention for publication once the answers have been provided to the few points raised

Reviewer #1 (Remarks to the Author):

I find that all of my comments (and comments from other reviewers) have been carefully addressed. I have no further comments. Thanks for this very nice contribution.

Reviewer #3 (Remarks to the Author):

The authors have taken the comments into account and responded to the various points raised with great accuracy and effort in explanations in the responses and modifications. Remarkable work!